# Mitigating Spurious Negative Pairs for Robust Industrial Anomaly Detection

**Hossein Mirzaei**[1], **Mojtaba Nafez**[2], **Jafar Habibi**[2], **Mohammad Sabokrou**[3], and **Mohammad Hossein Rohban**[2]

[1]École Polytechnique Fédérale de Lausanne (EPFL), Switzerland
[2]Sharif University of Technology, Iran
[3]Okinawa Institute of Science and Technology, Japan
rohban@sharif.edu, hossein.mirzaeisadeghlou@epfl.ch

## Abstract

Despite significant progress in Anomaly Detection (AD), the robustness of existing detection methods against adversarial attacks remains a challenge, compromising their reliability in critical real-world applications such as autonomous driving. This issue primarily arises from the AD setup, which assumes that training data is limited to a group of unlabeled normal samples, making the detectors vulnerable to adversarial anomaly samples during testing. Additionally, implementing adversarial training as a safeguard encounters difficulties, such as formulating an effective objective function without access to labels. An ideal objective function for adversarial training in AD should promote strong perturbations both within and between the normal and anomaly groups to maximize margin between normal and anomaly distribution. To address these issues, we first propose crafting a pseudo-anomaly group derived from normal group samples. Then, we demonstrate that adversarial training with contrastive loss could serve as an ideal objective function, as it creates both inter- and intra-group perturbations. However, we notice that spurious negative pairs compromise the conventional contrastive loss to achieve robust AD. Spurious negative pairs are those that should be closely mapped but are erroneously separated. These pairs introduce noise and misguide the direction of inter-group adversarial perturbations. To overcome the effect of spurious negative pairs, we define opposite pairs and adversarially pull them apart to strengthen inter-group perturbations. Experimental results demonstrate our superior performance in both clean and adversarial scenarios, with a **26.1%** improvement in robust detection across various challenging benchmark datasets. The implementation of our work is available at: `https://github.com/rohban-lab/COBRA`.

## 1 Introduction

Anomaly detection (AD), also referred to as one-class classification, aims to identify whether an input sample at the time of inference belongs to the normal In AD setup, the training data consists only of normal samples, and any additional information, such as labels, is unavailable Bendale and Boult (2015); Perera et al. (2021). Recently, a plethora of literature has emerged to address the problem of AD on images, demonstrating near-perfect performance on standard AD benchmarks Ruff et al. (2018); Tack et al. (2020); Bergman et al. (2020); Reiss et al. (2021); Bergmann et al. (2019); Krizhevsky et al. (2009). Nevertheless, these methods demonstrate a lack of robustness, especially when faced with adversarial attacks, as they encounter substantial performance deterioration when faced with such scenarios Azizmalayeri et al. (2022); Lo et al. (2022); Chen et al. (2020a); Shao et al. (2020; 2022); Béthune et al. (2023); Goodge et al. (2021); Chen et al. (2021a). This is due to the absence of anomaly samples in the training data, which results in insufficient exposure to adversarial perturbations on anomalous patterns during training. This shortcoming would make the model vulnerable to adversarial attacks on anomaly samples during inference Chen et al. (2020a; 2021a).

Numerous defense strategies have been developed to enhance the robustness of deep neural networks, with adversarial training emerging as a potential solution Bai et al. (2021); Madry et al. (2017). However, its application to AD is not straightforward, as it is primarily developed for multi-class and labeled setups. Motivated by this, we propose generating pseudo-anomaly group samples by applying hard augmentations to facilitate practical adversarial training in an anomaly detection (AD) setup. This process involves shifting normal training data to ensure that the shifted samples do not belong to the normal group, measured by their likelihood using a novel thresholding approach Glodek et al. (2013). We refer to the relationship between a normal sample and its transformed version as *opposite pairs*.

Given the availability of two groups—crafted anomaly and normal samples—during training, defining a loss function to incorporate them into the adversarial training presents a new challenge. Since the objective of test-time adversarial attacks is to manipulate normal samples to be confused with anomalies and vice versa, the optimal objective function should maximize the margin between the distributions of normal and anomaly samples in the learned embedding space while also achieving compact representations for each group. This can be adversarially accomplished by crafting strong intra- and inter-group perturbations Chen et al. (2021b); Cheng et al. (2023); Guo and Zhang (2021).

It has been demonstrated that Contrastive Learning (CL) Chen et al. (2020b); He et al. (2020) is more effective for AD compared to existing objective functions Guo et al. (2024); Reiss and Hoshen (2021); Tack et al. (2020). One can propose adversarial training with CL to develop a robust AD method. However, we noticed that adversarial training with the CL loss function falls short of achieving robust AD (see Table 6). The CL objective aims to bring positive pairs closer together and push negative pairs further apart. Positive pairs are constructed by applying light transformations to each instance, while any two instances in the training data are treated as negative pairs. We refer to negative pairs within the same group (normal-normal or anomaly-anomaly) as *spurious negative pairs*. These spurious negative pairs weaken the effectiveness of adversarial CL by misdirecting inter-group perturbations, thereby reduces the margin between groups Chen et al. (2021b).

To address this, we propose COBRA (anomaly-aware **CO**ntrastive-**B**ased approach for **R**obust **AD**), a new method that mitigates the effect of spurious negative pairs to learn effective perturbations. COBRA strategically utilizes opposite pairs, exclusively formed between normal and anomaly groups, ensuring they do not intersect with spurious negatives. This approach strengthens inter-group perturbations by emphasizing these opposite pairs in the loss function, thereby increasing the margin between groups. During training, the model adversarially targets positive pairs to push them together and opposite pairs to pull them apart. This simulates a wide range of adversarial perturbations covering inter- and intra-set variations, resulting in a robust anomaly detector.

**Contribution.** COBRA introduces a simple yet effective approach to generate anomaly samples and a novel loss function to establish a robust detection boundary. We evaluate COBRA in both adversarial and clean settings, where test samples are benign. In the adversarial scenario, we employ numerous strong attacks for robustness evaluation, including PGD-1000 Madry et al. (2017), AutoAttack Croce and Hein (2020), and Adaptive AutoAttack Liu et al. (2022). The results show that COBRA, without using any additional datasets or pretrained models, significantly outperforms existing methods in adversarial settings, achieving a 26.1% improvement in AUROC and competitive results in standard settings. Our experiments span various datasets, including large and real-world datasets such as Autonomous Driving Cordts et al. (2016), ImageNet Deng et al. (2009), MVTecAD Bergmann et al. (2019), and ISIC Codella et al. (2019), demonstrating COBRA's practical applicability. Additionally, we conducted ablation studies to examine the impact of various COBRA components, specifically our pseudo-anomaly generation strategy and the introduced adversarial training method.

## 2 PRELIMINARIES

**Anomaly Detection.** Outlier detection is categorized into different areas, such as AD and Out-of-Distribution (OOD) detection, depending on the availability of normal set samples' labels. An AD method decides whether $x$ belongs to the normal or anomaly set by assigning an anomaly score $A(x; f)$ using model f. Samples with an anomaly score higher than a pre-assumed threshold are predicted as anomalies, and vice versa Yang et al. (2021); Ruff et al. (2021). It is important to note that extending OOD detection methods to an AD setup is not feasible, as they rely on labeled

normal data for feature extraction. This highlights the need for robust AD methods. Azizmalayeri et al. (2022); Chen et al. (2021a); Kong and Ramanan (2021); Han et al. (2022).

**Adversarial Robustness of Anomaly Detectors.** An adversarial attack is a malicious attempt to perturb a data sample $x$ with an associated label $y$ into a new sample $x^*$ that maximizes the loss function $\ell(x^*; y)$. Additionally, an upper limit of $\epsilon$ confines the $l_p$ norm of the adversarial noise to prevent semantic alterations. Specifically, an adversarial example $x^*$ must satisfy the following equations: $x^* = \arg\max_{x': \|x-x'\|_p \leq \epsilon} \ell(x'; y)$ A prevalent and effective attack method is the Projected Gradient Descent (PGD) technique Madry et al. (2017), which entails iteratively maximizing the loss function by advancing towards the gradient sign of $\ell(x^*; y)$, employing a designated step size $\alpha$. To adapt adversarial attacks for AD, instead of maximizing the loss value, we aim to increase $A(x, f)$ if $x$ belongs to normal group and decrease it otherwise. The formulation of the attack would be: $x_0^* = x, \quad x_{t+1}^* = x_t^* + y \cdot \alpha \cdot \text{sign}\left(\nabla_x A(x_t^*; f_\theta)\right), \ x^* = x_k^*$. Here $k$ is the number of attack steps, $y = +1$ for normal samples and $y = -1$ for anomaly samples. The same setting is applied to other attacks in our study.

**Auxiliary Anomaly Sample Crafting.** CSI Tack et al. (2020) and CPAD Li et al. (2021) propose using fixed hard augmentation to create auxiliary samples. Specifically, CSI relies on Rotation, while CPAD considers CutPaste as a pseudo-anomaly. The GOE Kirchheim and Ortmeier (2022) method employs a pretrained GAN on ImageNet-1K to craft anomalies by targeting low-density areas. FITYM Mirzaei et al. (2022) employed an underdeveloped diffusion as a generator. Dream-OOD Du et al. (2023) uses both image and text domains to learn visual representations of normal instances in an embedding space of a pretrained stable diffusion Rombach et al. (2022) model trained on 5 billion data (e.g. LAION Schuhmann et al. (2022)). On the other hand, VOS Du et al. (2022) generates anomaly embeddings instead of image data. Details about each mentioned method can be found in C.

## 3 METHOD

**Motivation.** Adversarial training is one of the most promising approaches to enhance the robustness of deep neural networks. However, applying this technique to AD poses a significant challenge, as only a single concept class—the normal distribution—is available during training. A common approach to address this limitation is to incorporate an auxiliary anomaly dataset to improve robustness Azizmalayeri et al. (2022); Chen et al. (2021a; 2020a); Mirzaei et al. (2024a). However, leveraging such datasets is both costly and challenging, primarily due to the need for preprocessing and filtering out normal samples, which could otherwise provide misleading information to the detector. Moreover, the use of additional anomaly data can bias the model towards specific anomaly samples, reducing its generalizability to unseen anomalies Ming et al. (2022).

To overcome these limitations, we propose a simple yet effective method to craft anomaly samples directly from the normal data, thus eliminating the need for external anomaly datasets. Our approach involves applying hard augmentations (e.g., severe distortions) to normal samples, effectively pushing them towards the anomaly distribution. Importantly, prior work has demonstrated that the most effective anomalies for training are those that are closely related to the normal distribution, often referred to as "near anomaly samples" Ming et al. (2022); Mirzaei et al. (2022); Chen et al. (2021a). Our method satisfies this proximity, as the crafted anomalies maintain stylistic similarities with the normal samples due to their generation through augmentations. To ensure that the crafted anomalies are sufficiently distinct from the normal distribution, we introduce a thresholding mechanism to filter out false anomalies. Implementing this mechanism requires a model that accurately captures the distribution of normal data. However, in the AD setup, the training data is limited to a single semantic class (e.g., images of cars) without any supplementary information, posing a challenge for building such a model. To overcome this, we employ a self-supervised approach to extract meaningful representations from the normal data. Inspired by representative studies in AD Golan and El-Yaniv (2018); Hendrycks et al. (2019a); Tack et al. (2020), which demonstrate that using a $k-$class classifier to predict data transformations is an effective method for representation learning in one-class classification, we adopt this approach. We leverage the embeddings learned by the classifier to compute the likelihood of test samples and define a threshold for filtering out false anomalies.

Subsequently, we explore potential objective functions for adversarial training, focusing on CL given its recent success in AD tasks Guo et al. (2024). However, we observe that employing standard CL in adversarial training may not yield optimal results. This stems from the fact that, when training on a

dataset containing both normal and crafted anomaly samples, CL forms positive and negative pairs in a way that may compromise the margin between the normal and anomaly distributions. Specifically, CL seeks to uniformly repel negative pairs from each other, defined as all pairs except those that are augmentations of each other Chen et al. (2020c). Consequently, the negative pairs include normal-normal, anomaly-anomaly, and normal-anomaly pairs. Increasing the distance between normal-normal and anomaly-anomaly pairs may inadvertently reduce the separation between normal and anomaly sets, thus undermining robust detection performance. In other words, standard CL does not effectively enhance the inter-set margin needed to improve robustness. To address this, we design a novel objective function that explicitly maximizes the margin between the normal and anomaly groups to improve inter-group perturbation.

**Outline.** Existing AD methods experience dramatic performance decrease under adversarial attack. To address this, we propose COBRA, a novel method that integrates a distribution-aware transformation for generating psudo-anomaly samples, coupled with a novel objective function for adversarial training. In the subsequent sections, we will delve into each component in detail, outlining the mechanisms and advantages of our approach.

## 3.1 DISTRIBUTION AWARE HARD TRANSFORMATION

**Anomaly Crafting Strategy.** Previous works have demonstrated the effectiveness of leveraging an auxiliary random dataset as an additional source of anomaly data during the training phase for AD Hendrycks et al. (2018); Tao et al. (2023); Du et al. (2022; 2023); Zhang et al. (2017); Mirzaei et al. (2022); Kirchheim and Ortmeier (2022). However, this technique significantly depends on the diversity and distribution distance of the auxiliary dataset used for training. This limitation significantly hinders the use of this technique in areas like medical imaging, where real anomalies are scarce and difficult to obtain. Moreover, they lack any threshold for dropping incorrectly crafted anomalies (those that still belong to the normal group). Addressing this limitation, our approach introduces a novel method that employs a series of hard transformations $T = \{T_i\}_{i=1}^k$ to generate anomaly samples from normal data. We strategically distort normal images and by using a predetermined threshold ensures that the synthetically created samples significantly diverge from the normal distribution. We used a set of hard transformations including Jigsaw Noroozi and Favaro (2016), Random Erasing Zhong et al. (2020), CutPaste Ghiasi et al. (2020), Rotation, Extreme Blurring, Intense Random Cropping, Noise Injection Akbiyik (2019), and Extreme Cropping, Mixup Hongyi Zhang (2018), Cutout DeVries and Taylor (2017), CutMix Yun et al. (2019), Elastic transform and etc. Each one has been shown to be harmful for preserving semantics in previous studies Tack et al. (2020); Sohn et al. (2020); Park and Darrell (2020); de Haan and Löwe (2021); Kalantidis et al. (2020a); Li et al. (2021); Sinha et al. (2021); Kalantidis et al. (2020b); Miyai et al. (2023); Zhang et al. (2024); Chen et al. (2021c). For more details, please see Appendix D.1.

**Threshold Computing.** First, we train a transformation predictor model that captures the distribution of normal samples through the classification of various augmentations. To achieve this, we create a synthetic dataset with $k$ classes, denoted as $\{D_{\text{train}}^{T_i}\}_{i=1}^k$, by applying $k$ different hard transformations $T_i$ to each of the $n$ samples in the normal training set $D_{\text{train}}$. Then, we train a $k$-class classifier on this synthetic dataset to leverage its knowledge for threshold computation. Using the classifier as a feature extractor, denoted as $C$, we extract embeddings of the normal training samples to create an embedding set: $e_{\text{train}} = \left\{C(D_{\text{train}}^i)\right\}_{i=1}^n$. Next, we fit a Gaussian Mixture Model (GMM) to the training data embeddings $e_{\text{train}}$, as this is a well-established approach in the literature Cohen and Avidan (2021); Du et al. (2022). The likelihood for each sample is computed, and the **p-value** for test samples is calculated based on the empirical distribution of likelihoods from the normal training samples. The threshold $\lambda$ is set at a default significance level of 0.05, such that samples with p-values below this threshold are considered anomalies. An ablation study on the significance level, as well as an analysis of $C$, are provided in Appendix D and Appendix E, respectively.

**Opposite Pairs with Pseudo-Anomaly Samples.** For each normal sample, we randomly select a subset of transformations $T$, containing at least two transformation. These transformations are applied in a randomized sequence to the sample, producing $x' = T_{i_m}(\ldots T_{i_1}(x))$ where $m < k$. We then get its embedding and calculate the likelihood $pr(C(x'))$. Finally, this likelihood is compared against the computed threshold $\lambda$. Samples exceeding this threshold iteratively repeat this process until deemed an anomaly. We represent our proposed strategy for anomaly crafting with the notation $\Upsilon(x)$. Before each step of training, given a batch of normal samples denoted by $\mathcal{B}_{\text{normal}} = \{x^j\}_{j=1}^b$,

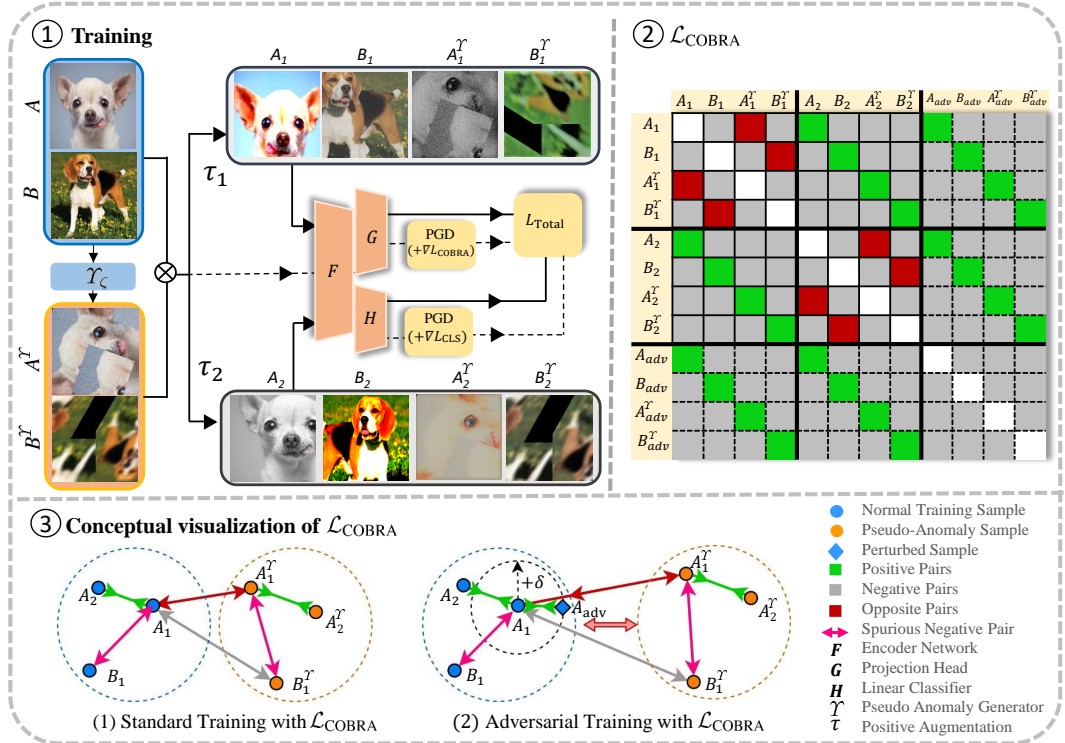

Figure 1: ①: Given a training batch that includes normal group samples $A$ and $B$, we create a anomaly group using our proposed transformation $\Upsilon_\lambda$. These samples are paired as opposite pairs (e.g., $A$ and $A^\Upsilon$) and subjected to $\tau_1$ and $\tau_2$ to form batches of positive pairs. Adversarial training is performed with a loss function combining $L_{\text{CLS}}$ and $L_{\text{COBRA}}$, where $L_{\text{COBRA}}$ treats adversarial examples as positive pairs for the corresponding sample. ②, ③: The illustrations demonstrate how $L_{\text{COBRA}}$ enhances adversarial training by explicitly increasing similarities within positive pairs and decreasing similarity for opposite pairs, thus creating strong inter- and intra-group perturbations. Targeting opposite pairs instead of all negatives diminishes the effect of spurious negative pairs (e.g., $(A_1, B_1)$), leading to stronger inter-group perturbations and enlarging the margin between distributions for normal and anomaly groups. A detailed algorithmic of COBRA is provided in A.

we create a batch of anomaly samples $\mathcal{B}_{\text{p-anomaly}} = \{x^j\}_{j=b+1}^{2b}$. Where $x^{b+i} = \Upsilon(x^i)$ signifies an anomaly sample derived from applying a transformation $\Upsilon$ to a normal sample $x^i$, and $(x^i, x^{b+i})$ are considered as symmetrically opposite pairs. During training, the notation $\Upsilon(x)$ has some minor differences, where $\Upsilon(x^{b+i})$ is considered as $x^i$.

## 3.2 ADVERSARIAL TRAINING WITH ANOMALY-AWARE CL

**Conventional Contrastive Loss.** In the conventional CL paradigm, each instance $x$ within a batch transforms into two positive views, $(x_1, x_2)$, via a random selection of positive augmentations $\tau_1, \tau_2$ from a predefined set $\mathcal{T}$. The set of positive pairs corresponding to a sample $x$ is denoted as $P(x_1) = \{x_2\}$ and $P(x_2) = \{x_1\}$. CL then defines negative pairs $N(x)$ for sample $x$ as the other samples' augmented views. By denoting the current batch as $B$, we achieve: $N(x) = \{\tau_1(x') : x' \in B \setminus \{x\}\} \cup \{\tau_2(x') : x' \in B \setminus \{x\}\}$. These views are processed through a target network to obtain projected features, symbolized as $z = \mathcal{G}(\mathcal{F}(x))$, where $\mathcal{F}$ signifies the feature encoder and $\mathcal{G}$ the projection head. For simplification, $f(.)$ substitutes $\mathcal{G}(\mathcal{F}(.))$. The conventional NT-Xent Chen et al. (2020b) loss is articulated as:

$$\mathcal{L}_{\text{CL}}(x) = -\sum_{i=1}^{2} \sum_{x_j \in P(x_i)} \log \frac{\exp(\text{sim}(f(x_i), f(x_j))/t)}{\sum_{x_k \in P(x_i) \cup N(x)} \exp(\text{sim}(f(x_i), f(x_k))/t)}. \tag{1}$$

where $\text{sim}(\cdot, \cdot)$ denotes the cosine similarity function, $t$ is the temperature parameter, $\text{P}(x_i)$ is the set of positive pairs for $x_i$, and $\text{N}(x_i)$ is the set of negative pairs for $x_i$.

**Addressing Spurious Negative Pairs with $\mathcal{L}_{\text{COBRA}}$.** CL aims to pull positive pairs closer to each other and push negative pairs away from each other. Consider defining the current batch of samples for CL as the concatenation of two groups: normal and anomaly samples, $\mathcal{B} = \{\mathcal{B}_{\text{normal}} \cup \mathcal{B}_{\text{p-anomaly}}\}$. Due to the definition of negative pairs in CL, each sample in $\mathcal{B}$ includes both inter-group and intra-group relations as negative pairs. Intra-group negative pairs, i.e., normal-normal and anomaly-anomaly pairs, are considered spurious negative pairs. Distancing spurious negative pairs is counterproductive to our objective, which is to maximize the discriminative margin between normal and anomaly groups. Specifically, in the scenario of adversarial training for robust AD with $\mathcal{L}_{\text{CL}}$, spurious negative pairs misdirect inter-group adversarial perturbation. As a result, we aim to precisely target those negative pairs that definitely belong to separate groups—what we refer to as opposite pairs. By focusing on these pairs, we aim to induce stronger perturbations that significantly enhance the discriminative margin between the normal and anomaly groups by proposing $\mathcal{L}_{\text{COBRA}}$.

$$\mathcal{L}_{\text{COBRA}}(x) = -\sum_{i=1}^{2} \sum_{x_j \in \text{P}(x_i)} \log \frac{\exp(\text{sim}(f(x_i), f(x_j))/t) - \exp(\text{sim}(f(x_i), f(\Upsilon(x)))/t)}{\sum_{x_k \in \text{P}(x_i) \cup \text{N}(x)} \exp(\text{sim}(f(x_i), f(x_k))/t)}, \quad (2)$$

Note that $\Upsilon(x)$ could also be replaced by $\Upsilon(x_i)$, as both $x$ and $x_i$ are positive pairs and share similar semantics. Applying a hard transformation to either would result in a comparable hard transformation. The intuition behind $\mathcal{L}_{\text{COBRA}}$ is that the representations of the corresponding positive views $x_1$ and $x_2$ should be similar (analogous to the $\mathcal{L}_{\text{CL}}$ loss function), leading to compact representations for each group. Meanwhile, the representation of $x$ should be distinctly different from its counterpart representation $\Upsilon(x)$, resulting in a high margin between the two groups. A conceptual visualization of $\mathcal{L}_{\text{COBRA}}$ is provided in Figure 1. It is important to highlight that the limitations of CL in AD are apparent in adversarial scenarios. This stems from the fact that adversarial training requires a higher degree of data complexity Schmidt et al. (2018); Stutz et al. (2019) compared to clean settings, necessitating a broad range of strong perturbations to achieve robust anomaly detection. One can say $\mathcal{L}_{\text{COBRA}}$ consists of two terms, $\mathcal{L}_{\text{CL}}$ and $\mathcal{L}_{\text{Opposite}}$, where

$$\mathcal{L}_{\text{Opposite}}(x) = -\sum_{i=1}^{m} \sum_{x_j \in \text{P}(x_i)} \log \frac{-\exp(\text{sim}(f(x_i), f(\Upsilon(x))/t)}{\sum_{x_k \in \text{P}(x_i) \cup \text{N}(x_i)} \exp(\text{sim}(f(x_i), f(x_k))/t)}, \quad (3)$$

To enhance our model's ability to distinguish between normal and anomaly groups, we employ a fully connected layer followed by softmax activation for binary classification, denoted as $\mathcal{H}$, following the $\mathcal{F}$. For the classification loss, $\mathcal{L}_{\text{CLS}}(x)$, we define a label $y$ corresponding to the batch $\mathcal{B}$, where '0' is assigned as the label for normal samples and '1' for pseudo anomaly samples. Our final loss function, $\mathcal{L}_{\text{COBRA}}$, is thus formulated as: $\mathcal{L} = \sum_{i=1}^{2b} \mathcal{L}_{\text{COBRA}}(\mathcal{F}, \mathcal{G}; \mathcal{B}^i) + \mathcal{L}_{\text{CLS}}(\mathcal{F}, \mathcal{H}; \mathcal{B}^i, y^i)$

**Adversarial Training Step.** For $\mathcal{L}_{\text{COBRA}}$, given an input sample $x$ from the batch $\mathcal{B}$, an adversarial example $x_{adv}$ is generated by introducing a perturbation $\delta^*$, optimized to maximize our final loss: $\delta^* = \arg\max_{\|\delta\|_\infty \le \epsilon} \mathcal{L}(x + \delta, y), \quad x_{adv} = x + \delta^*$. Then, adversarial examples are used in the training process alongside the original examples. Specifically, for $\mathcal{L}_{\text{COBRA}}$, we consider them as another positive view of each sample and aim to align each sample with its perturbed version, i.e., $P(x_i) \leftarrow P(x_i) \cup \{x_{adv}\}$. The adversarial training objective as a min-max problem, optimizing the model parameters $\theta$ to minimize the expected loss over both clean and adversarial examples:

$$\min_\theta \mathbb{E}_{(x,y) \in \mathcal{B}} \left[ \max_{\|\delta\|_\infty \le \epsilon} \mathcal{L}(x + \delta, y; \theta) \right].$$

Morever, the stability of the $\mathcal{L}_{\text{COBRA}}$ can be observed in both clean and adversarial training scenarios, as illustrated in the Appendix F.

**Anomaly Score for Evaluation.** For evaluating anomalies, we leverage the representation learned by $\mathcal{F}$ to compute the anomaly score, based on the similarity between test samples and normal training samples in the embedding space. The anomaly score $A(X)$ for a test sample $x$ is defined as: $-\max_{x^i \in D_{\text{train}}} \{sim(f(\mathbf{x}), f(\mathbf{x}^i))\}$, This scoring mechanism takes advantage of the contrastive

Table 1: Performance of AD methods on MVTecAD dataset under clean evaluation and PGD-1000 adversarial attack with $\epsilon = \frac{2}{255}$, measured by AUROC (%). The best results are emphasized in bold format in each row. The table cells denote results in the 'Clean / PGD-1000' format.

*These works incorporated adversarial training into their proposed AD methods.

| Category | Method | | | | | | | | |
|---|---|---|---|---|---|---|---|---|---|
| | CSI | Transformaly | PatchCore | ReContrast | DRÆM | PrincipaLS* | OCSDF* | ZARND* | COBRA (Ours) |
| Carpet | 50.2 / 11.1 | 95.5 / 0.0 | 98.7 / 18.4 | 99.8 / 9.4 | 97.0 / 0.0 | 54.8 / 33.6 | 56.1 / 12.6 | **85.9** / 66.6 | 60.7 / **84.9** |
| Grid | 71.2 / 8.3 | 84.2 / 7.8 | 98.2 / 11.7 | 100.0 / 19.8 | 99.9 / 2.7 | 72.1 / 30.4 | 61.7 / 17.3 | 75.7 / 31.1 | **100.0** / **99.5** |
| Leather | 70.9 / 0.4 | 99.9 / 4.1 | 100.0 / 10.5 | 100.0 / 3.4 | **100.0** / **0.0** | 73.2 / 26.5 | 61.4 / 13.7 | 65.0 / 14.1 | 97.4 / **91.7** |
| Tile | 67.8 / 7.2 | 97.1 / 2.0 | 98.7 / 4.6 | 99.8 / 2.4 | 99.6 / 0.0 | 58.7 / 26.3 | 54.3 / 10.1 | 53.6 / 3.9 | **98.8** / **78.2** |
| Wood | 71.3 / 6.0 | 98.5 / 0.0 | 99.2 / 3.8 | 99.0 / 1.6 | 99.1 / 1.8 | 67.3 / 31.2 | 63.9 / 3.7 | 58.4 / 17.5 | **96.4** / **73.7** |
| Bottle | 69.4 / 1.2 | 99.4 / 5.1 | 100.0 / 9.4 | 100.0 / 6.8 | 99.2 / 2.1 | 72.1 / 29.4 | 59.8 / 9.1 | 79.9 / 54.9 | **100.0** / **88.8** |
| Cable | 66.5 / 7.9 | 81.5 / 0.1 | 99.5 / 4.3 | 99.8 / 3.3 | 91.8 / 1.9 | 63.9 / 26.2 | 61.6 / 6.3 | 68.5 / 30.0 | **92.4** / **74.8** |
| Capsule | 51.6 / 6.8 | 76.0 / 0.0 | 98.1 / 3.1 | 97.7 / 2.7 | 98.5 / 0.0 | 56.8 / 18.4 | 51.9 / 1.9 | 69.3 / 26.2 | **75.9** / **55.8** |
| HazelNut | 66.7 / 0.0 | 89.8 / 0.4 | 100.0 / 7.8 | 100.0 / 4.1 | **100.0** / 0.8 | 64.8 / 21.7 | 54.2 / 4.7 | 73.2 /24.0 | **96.3** / **74.7** |
| MetalNut | 65.8 / 0.7 | 90.9 / 6.2 | 100.0 / 4.8 | 100.0 / 3.7 | 98.7 / 0.6 | 61.6 / 19.4 | 59.5 / 3.5 | 43.1 / 1.3 | **96.8** / **78.1** |
| Pill | 48.3 / 3.1 | 83.7 / 2.0 | 96.6 / 2.0 | 98.6 / 1.8 | 98.9 / 0.0 | 52.5 / 9.4 | 57.4 / 0.7 | **84.0** / 42.9 | 57.7 / **53.2** |
| Screw | 51.7 / 0.0 | 73.3 / 0.4 | 98.1 / 0.0 | / 98.0 / 3.8 | 93.9 / 0.0 | 57.6 / 3.7 | 55.0 / 0.6 | 84.7 / 21.4 | 74.2 / **36.4** |
| Toothbrush | 75.3 / 1.3 | 90.8 / 0.9 | 100.0 / 6.9 | 100.0 / 6.7 | 100.0 / 0.0 | 70.8 / 28.2 | 60.1 / 6.4 | 65.9 / 22.3 | **100.0** / **75.5** |
| Transistor | 61.7 / 9.8 | 76.4 / 2.5 | **100.0** / 7.8 | 99.7 / 8.1 | 93.1 / 5.3 | 60.1 / 26.3 | 58.7 / 4.5 | 86.5 / 46.3 | 91.0 / **69.2** |
| Zipper | 68.2 / 5.4 | 90.7 / 0.8 | 99.4 / 13.6 | 99.5 / 13.7 | **100.0** / 4.7 | 67.9 / 35.7/ | 65.6 / 9.8 | 82.2 /48.6 | 99.2 / **92.5** |
| **Average** | 63.8 / 4.6 | 88.5 / 2.2 | 99.1 / 7.2 | **99.5** / 6.1 | 98.0 / 1.3 | 63.6 / 24.4 | 58.7 / 7.0 | 71.6/ 30.1 | 89.1 / **75.1** |

training framework, ensuring that normal test samples exhibit higher similarity scores in comparison to anomaly test samples. Consequently, the anomaly score for anomaly test samples will be notably higher than for normal test samples, enabling robust AD. Alternative anomaly scores have been explored in the appendix E.

Table 2: Performance of AD methods on various datasets under clean evaluation and PGD-1000. For the experiments across **all tables**, adversarial attacks were considered, using $\epsilon = \frac{4}{255}$ for low-resolution images and $\epsilon = \frac{2}{255}$ for high-resolution images, measured by AUROC (%). The table cells denote results in the 'Clean / PGD' format. Experiments performed in the one-class AD setup.

*These works incorporated adversarial training into their proposed AD methods.

| | Dataset | Method | | | | | | | | | |
|---|---|---|---|---|---|---|---|---|---|---|---|
| | | DeepSVDD | CSI | MSAD | Transformaly | PatchCore | PrincipaLS* | OCSDF* | APAE* | ZARND* | COBRA (Ours) |
| Low Res | CIFAR10 | 64.8 / 8.7 | 94.3 / 10.6 | 97.2 / 4.8 | **98.3** / 3.7 | 68.3 / 3.9 | 58.3 / 33.2 | 58.7 / 31.3 | 56.3 / 2.2 | 89.7 / 56.0 | 83.7 / **62.3** |
| | CIFAR100 | 67.0 / 3.6 | 89.6 / 11.9 | 96.4 / 8.4 | **97.3** / 9.4 | 66.8 / 4.3 | 51.9 / 26.2 | 50.2 / 23.5 | 53.1 / 4.1 | 88.4 / 47.6 | 76.9 / **51.7** |
| | MNIST | 94.8 / 8.2 | 93.8 / 3.4 | 96.0 / 3.2 | 94.8 / 7.9 | 83.2 / 2.6 | 97.8 / 83.1 | 96.1 / 68.9 | 93.4 / 34.7 | 99.0 / 91.2 | 92.8 / **96.4** |
| | FMnist | 94.5 / 7.9 | 92.7 / 5.8 | 94.2 / 6.6 | 94.4 / 7.4 | 77.4 / 5.5 | 92.5 / 69.2 | 91.8 / 64.9 | 88.3 / 19.5 | **95.0** / 82.3 | 93.1 / **89.6** |
| | SVHN | 60.3 / 1.5 | **96.8** / 3.1 | 58.3 / 0.2 | 56.9 / 0.9 | 52.1 / 2.1 | 63.0 / 11.2 | 58.1 / 9.7 | 52.6 / 1.4 | 53.5 / 9.6 | 89.3 / **58.2** |
| High Res | ImageNet | 56.4 / 4.0 | 91.6 / 5.6 | 98.9 / 2.6 | **99.0** / 2.9 | 67.6 / 2.5 | 56.2 / 28.3 | 55.3 / 25.8 | 58.3 / 2.1 | 96.4 / 27.4 | 85.2 / **57.0** |
| | VisA | 53.6 / 1.8 | 62.5 / 0.3 | 84.1 / 4.6 | 85.5 / 0.0 | **95.1** / 2.7 | 57.3 / 16.1 | 53.0 / 13.9 | 67.2 / 9.1 | 71.8 / 24.9 | 75.2 / **73.8** |
| | CityScapes | 59.7 / 2.7 | 68.9 / 0.1 | 86.5 / 2.9 | 87.4 / 4.5 | **76.2** / 6.1 | 60.3 / 24.2 | 59.6 / 20.1 | 63.0 / 3.6 | 75.9 / 28.6 | 81.7 **56.2** |
| | DAGM | 57.3 / 2.7 | 74.5 / 1.6 | 73.8 / 0.0 | 81.4 / 0.5 | **93.6** / 1.9 | 59.2 / 24.8 | 57.6 / 20.3 | 54.5 / 13.8 | 64.5 / 17.2 | 82.4 / **56.8** |
| | ISIC2018 | 64.1 / 0.3 | 71.2 / 0.0 | 76.7 / 3.4 | **86.6** / 3.9 | 78.9 / 0.0 | 61.7 / 26.5 | 64.0 / 18.6 | 67.2 / 8.5 | 70.2/ 14.6 | 81.3 / **56.1** |
| | **Average** | 67.3 / 4.1 | 83.6 / 4.2 | 86.2 / 3.7 | **88.1** / 4.1 | 75.9 / 3.1 | 65.8 / 34.3 | 64.5 / 29.7 | 65.4 / 9.9 | 80.4 /39.7 | 84.1 / **65.8** |

## 4  EXPERIMENTS

In this section, we verify the effectiveness of COBRA in robust AD with several benchmark datasets, encompassing those that are large-scale and real-world. Evaluation is conducted to assess existing AD methods, including both clean and adversarially trained methods, as well as our own method, under both clean and various adversarial attack scenarios. Table 1 provides a comparative analysis in a one-class setting on the MVTecAD dataset, a challenging real-world benchmark in AD. Additional

Table 3: Performance of AD methods under clean evaluation and PGD-1000, measured by AUROC (%). Experiments performed in the unlabeled multi-class AD setup.

[*]These works incorporated adversarial training into their proposed AD methods.

| In | Out | Method | | | | | | |
|---|---|---|---|---|---|---|---|---|
| | | MSAD | Transformaly | PrincipaLS[*] | OCSDF[*] | APAE[*] | ZARND[*] | COBRA (*Ours*) |
| CIFAR10 | CIFAR100 | 76.9 / 0.4 | 88.7 / 0.0 | 54.8 / 14.6 | 51.0 / 12.8 | 53.6 / 1.2 | 76.6 / 34.1 | 76.0 / **63.3** |
| | SVHN | 94.6 / 0.0 | 98.2 / 1.2 | 72.1 / 23.6 | 67.7 / 18.8 | 60.8 / 2.1 | 84.3 / 42.7 | 98.5 / **78.6** |
| | MNIST | 99.3 / 1.6 | 99.4 / 3.6 | 82.5 / 42.7 | 74.2 / 37.4 | 71.3 / 15.3 | 99.4 / 82.2 | 80.8 / **85.8** |
| | FMnist | 99.2 / 3.8 | 99.1 / 3.7 | 78.3 / 38.5 | 64.5 / 33.7 | 59.4 / 9.4 | 98.2 / 67.3 | 82.8 / **75.7** |
| | ImageNet | 83.7 / 0.0 | 92.8 / 0.8 | 55.3 / 12.3 | 52.8 / 10.2 | 56.1 / 0.3 | 71.5 / 28.4 | 85.5 / **53.1** |
| CIFAR100 | CIFAR10 | 61.4 / 0.0 | 82.5 / 0.3 | 47.6 / 8.1 | 51.1 / 6.3 | 50.5 / 0.7 | 64.6 / 21.2 | 48.7 / **27.5** |
| | SVHN | 86.6 / 2.7 | 94.7 / 2.6 | 66.3 / 13.2 | 58.7 / 9.2 | 58.1 / 1.1 | 70.0 / 26.8 | 93.2 / **49.4** |
| | MNIST | 97.4 / 3.5 | 98.8 / 0.8 | 80.4 / 30.4 | 76.4 / 28.9 | 74.7 / 11.8 | 87.0 / 30.4 | 77.8 / **54.1** |
| | FMnist | 96.5 / 0.9 | 98.4 / 5.2 | 72.7 / 18.7 | 62.8 / 14.3 | 60.9 / 9.7 | 97.3 / **76.3** | 58.2 / 32.9 |
| | ImageNet | 71.6 / 1.6 | 80.4 / 2.0 | 51.6 / 6.3 | 48.9 / 5.4 | 52.7 / 0.1 | 71.6 / 21.8 | 69.1 / **32.3** |

Table 4: Performance of COBRA on various datasets under clean evaluation and several adversarial attack, measured by AUROC (%). Experiments performed in the one-class AD setup.

| Dataset | Attack | | | | | | |
|---|---|---|---|---|---|---|---|
| | Clean | BlackBox | FGSM | CAA | $A^3$ | AutoAttack | PGD-1000 |
| CIFAR10 | 83.7 | 81.8 | 70.2 | 64.5 | 60.7 | 65.9 | 62.3 |
| CIFAR100 | 76.9 | 74.6 | 64.5 | 53.0 | 50.1 | 54.8 | 51.7 |
| FMnist | 93.1 | 92.9 | 90.7 | 91.6 | 90.8 | 87.4 | 89.6 |
| ImageNet | 85.2 | 82.0 | 71.4 | 53.6 | 61.8 | 59.4 | 57.0 |
| MVTecAD | 89.1 | 83.4 | 79.8 | 76.3 | 74.8 | 77.0 | 75.1 |
| VisA | 75.2 | 74.6 | 74.0 | 73.5 | 71.6 | 74.9 | 73.8 |

comparisons in one-class settings across other benchmarks are detailed in Table 2, while Table 3 showcases our method's superiority in unlabeled multi-class setting. Moreover, we demonstrate our method's robustness by evaluating it against several attacks presented in Table 4. Additional details on the adaptation of attacks and supplementary evaluation metrics can be found in K and N.

**Experimental Setup.** Our experiments were conducted in two categories: one-class and unlabeled multi-class anomaly detection (AD). In the one-class setup, considering a dataset $D$ with $M$ classes, experiments were conducted by treating each class in turn as the normal set and the other $M - 1$ classes as the anomaly set. This process was repeated for each class, and performance was averaged across all classes to report the overall detection performance. In the unlabeled multi-class setup, this setting incorporates another dataset $D'$, considering one dataset as the normal set and another as the anomaly set. We compared COBRA with PANDA Reiss et al. (2021), Transformaly Cohen and Avidan (2021), Patchcore Roth et al. (2021), CSI Tack et al. (2020), MSAD Reiss and Hoshen (2021), ReContrast Guo et al. (2024), and Draem Zavrtanik et al. (2021), as well as methods specifically proposed for robust AD, including ZARND Mirzaei et al. (2024b), PrincipaLS Lo et al. (2022), OCSDF Béthune et al. (2023), and APAE Goodge et al. (2021). Details about each mentioned method can be found in Appendix C.

**Evaluation Details.** To evaluate the methods' adversarial robustness, both normal and anomalous test samples will be subjected to end-to-end adversarial attacks targeting the methods' anomaly scores. We set the value of $\epsilon$ to $\frac{4}{255}$ for low-resolution datasets and to $\frac{2}{255}$ for high-resolution datasets. For the PGD attack, we set the number of steps $N$ to 1000, initializing the attack from 10 different random starting points for each trial to enhance the attack's effectiveness and coverage. Furthermore, to highlight COBRA's robust performance, we considered additional strong attacks, including AutoAttack (AA), Adaptive AutoAttack ($A^3$), and black-box attacks. Furthermore, to highlight COBRA's robust performance, we considered an additional range of simple to strong attacks, including black-box attacks Guo et al. (2019), FGSM attacks Goodfellow et al. (2014), CAA Mao et al. (2020), AutoAttack (AA), and Adaptive AutoAttack ($A^3$). Other methods' performance under AutoAttack can be found in Appendix G. Additionally, details on the model's evaluation under both $\ell_\infty$ and $\ell_2$ PGD attacks across varying epsilon values are presented in Appendix H.

**Implementation Details & Datasets** For obtaining the threshold $\lambda$, we utilized a from-scratch ResNet-18 as $C$ and trained on the created dataset for 100 epochs. For adversarial training, we use PGD-10 step and $\epsilon = \frac{4}{255}$. We employ ResNet-18 as the foundational encoder network, accompanied

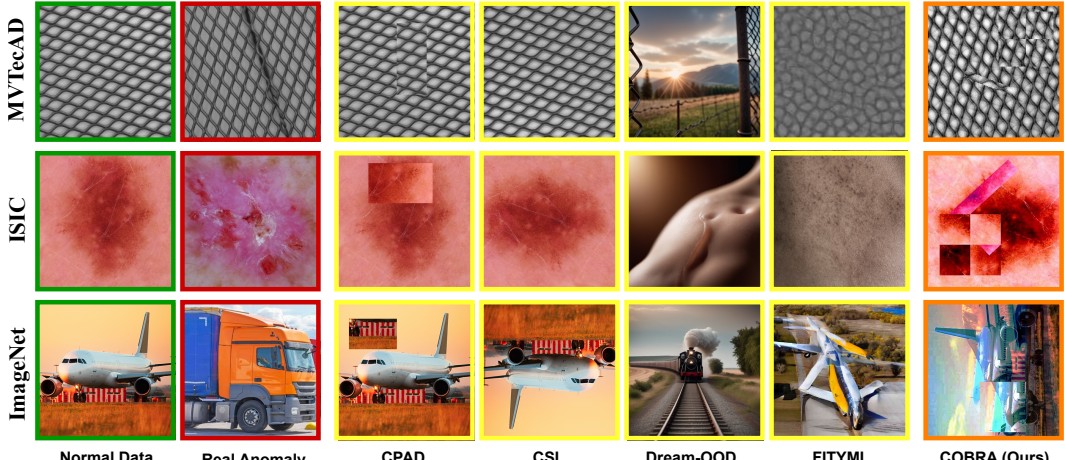

Figure 2: The figure highlights the challenges CPAD and CSI face in generating inncorrect anomalies due to the absence of a threshold. Techniques such as FITYMI and Dream-OOD, which generate anomalies from the embedding space of a pretrained model, typically lead to a loss of pixel-level detail and show biases towards the dataset used for pre-training (e.g., ImageNet). Such biases decrease their effectiveness on datasets not seen during pre-training, such as medical imaging datasets like ISIC. In contrast, COBRA efficiently crafts informative anomalies in the pixel space and utilizes a thresholding method to filter out incorrect anomalies, all **without** the need for any additional datasets.

Table 5: Comparison of COBRA with replacing alternative anomaly synthesis methods. *None* corresponds to a scenario where we neglect any pseudo-anomaly and adapt COBRA for that setting.

| Dataset | Anomaly Craft Strategy | | | | | | | |
|---------|------|------|------|------|------|--------|-----------|------|
| | *None* | CPAD | CSI | GOE | VOS | FITYMI | Dream-OOD | Ours |
| MVTecAD | 57.6 / 10.8 | 86.2 / 70.8 | 61.4 / 12.9 | 58.1 / 25.3 | 53.4 / 15.8 | 65.0 / 38.7 | 68.6 / 36.4 | **89.1 / 75.1** |
| ImageNet | 72.8 / 32.5 | 67.5 / 43.4 | 82.7 / 58.2 | 81.9 / 60.4 | 72.4 / 56.5 | 68.9 / 47.2 | **87.3 / 64.1** | 85.2 / 57.0 |
| CIFAR10 | 78.6 / 50.3 | 69.4 / 53.8 | 82.9 / 60.7 | **84.2** / 58.8 | 79.3 / 53.1 | 76.9 / 50.6 | 75.2 / 57.8 | 83.7 / **62.3** |
| FMnist | 82.4 / 71.7 | 86.3 / 78.5 | 89.5 / 82.6 | 73.9 / 64.1 | 68.2 / 61.9 | 71.7 / 62.0 | 76.4 / 68.5 | **93.1 / 89.6** |
| **Average** | 72.8 / 41.3 | 77.3 / 61.6 | 79.0 / 53.1 | 74.5 / 52.1 | 68.3 / 47.3 | 70.6 / 50.3 | 76.9 / 56.7 | **87.8 / 71.0** |

by an auxiliary head comprising a 2-layer multi-layer perceptron with a 128-dimensional embedding dimension. More details about the implementation can be found in Appendix I. COBRA is evaluated using challenging datasets that includes both high- and low-resolution images. The high-resolution dataset comprises MVTecAD Bergmann et al. (2019), VisA Zou et al. (2022), CityScapes Cordts et al. (2016), ImageNet Deng et al. (2009), ISIC2018 Codella et al. (2019), and DAGM Wieler et al. (2007), while the low-resolution dataset includes SVHN Goodfellow et al. (2013), FMNIST Xiao et al. (2017), CIFAR10, CIFAR100, and MNIST. Further details can be found in Appendix L.

**Analyzing Results.** The results presented underscore COBRA's effectiveness as an robust AD method. Remarkably, COBRA enhances the average robust detection performance across various datasets by up to **26.1%**, **without** relying on pre-trained models or extra datasets. This demonstrates COBRA's real-world applicability by enhancing robust performance on the MVTecAD dataset from **30.1%** to **75.1%**. COBRA's versatility is further highlighted by its general applicability to different AD scenarios, including one-class and unlabeled multi-class setups. Notably, in open-world applications where robustness is vital, a slight drop in clean performance is considered a worthwhile trade-off for enhanced robustness. Our results align with this perspective, achieving an average of **84.1%** in clean and **65.8%** in adversarial settings across various datasets. This performance surpasses methods like Transformaly Cohen and Avidan (2021), which, while achieving 88.2% in clean settings, significantly falls to 4.1% in adversarial scenarios. Furthermore, we replaced our adversarial training with clean training in the COBRA Pipeline. As expected, and in line with findings reported in the literature Tsipras et al. (2018), this resulted in decreased robust detection performance. However, it improved clean detection performance from an AUROC of 84.1% to **90.7%**.

Table 6: Comparison of the performance of COBRA with alternative loss functions versus $\mathcal{L}_{\text{COBRA}}$, in terms of effectiveness for the robust AD task.

| Dataset | Loss Function | | | | | |
|---------|---------------|---|---|---|---|---|
| | $\mathcal{L}_{\text{CLS}}$ | $\mathcal{L}_{\text{CL}}$ | $\mathcal{L}_{\text{SupCL}}$ | $\mathcal{L}_{\text{Opposite}}$ | $\mathcal{L}_{\text{COBRA}}$ | $\mathcal{L}_{\text{COBRA}} + \mathcal{L}_{\text{CLS}}$ |
| MVTecAD | 62.6 / 40.4 | 76.4 / 58.7 | 80.4 / 60.5 | 64.0 / 53.6 | 83.7 / 68.2 | **89.1 / 75.1** |
| ImageNet | 59.5 / 45.1 | 68.3 / 47.6 | 74.6 / 46.3 | 57.4 / 45.8 | 82.9 / 54.3 | **85.2 / 57.0** |
| CIFAR10 | 62.6 / 49.6 | 67.9 / 54.3 | 74.2 / 53.8 | 65.9 / 52.4 | 78.5 / 61.0 | **83.7 / 62.3** |
| FMnist | 82.0 / 78.4 | 88.3 / 82.5 | 91.5 / 83.2 | 80.4 / 73.5 | 92.8 / 87.6 | **93.1 / 89.6** |
| **Average** | 66.7 / 53.3 | 75.2 / 60.8 | 80.4 / 60.9 | 66.9 / 56.3 | 84.5 / 67.7 | **87.7 / 71.0** |

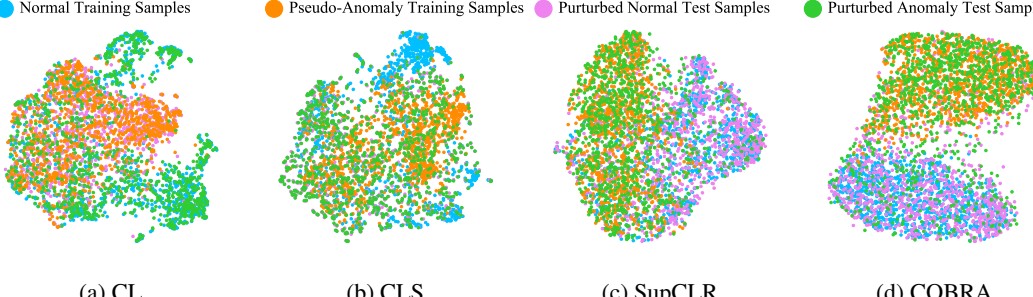

| | | | |
|---|---|---|---|
| (a) CL | (b) CLS | (c) SupCLR | (d) COBRA |

Figure 3: UMAP visualization of features extracted by the encoder $f$, trained with various loss functions on the CIFAR-10 dataset, is presented in a one-class setup with the 'Automobile' class designated as the normal set. For this particular experiment, all elements except the loss function remain constant to ensure a fair comparison.

## 5  ABLATION STUDY

**Pseudo-anomaly Generating Strategy.**   In order to demonstrate the superiority of our strategy for pseudo-anomaly sample crafting, with other modules fixed, we replaced ours with alternative methods. We provided a brief description of alternative methods in Section 3.1. The results, which are presented in Table 5, along with a visualization comparison of samples in Figure 2, show the superiority of our effective synthesizer method. Notably, our strategy, without using any extra data, outperforms Dream-OOD with billions of sample complexity by a margin of 15%. In Appendix D.1, we further evaluate the quality of our generated data.

**Adversarial Training Objective Function.**   We replaced our proposed loss function with various alternatives, such as classification (CLS), CL and Supervised CL. The results, detailed in Table 6, reveal that the COBRA loss function, by generating challenging intra- and inter-group adversarial examples during training, surpasses other alternatives significantly. $L_{\text{COBRA}}$ outperforms CLS by increasing normal distribution compactness provided by intra-group perturbations, and outperforms CL and SupCL by considering opposite pairs for increasing margins, as illustrated in Figure 3. Our performance outperforms other loss functions by 11%. Additional ablation studies, experimental results including error bars, limitations, and qualitative visualizations are provided in the Appendix.

## 6  CONCLUSION

In conclusion, our work introduces COBRA, a novel and effective approach for enhancing AD methods' robustness against adversarial attacks. By leveraging a novel loss function inspired by contrastive learning and strategically crafting informative anomaly samples, COBRA achieves superior detection performance under both clean and adversarial evaluation conditions. We verify COBRA through comprehensive ablation experiments on its different components. Moreover, our extensive experiments across multiple challenging datasets, as well as under various strong attacks, confirm our method's effectiveness, setting a new benchmark for future research in reliable AD.

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

# A ALGORITHM BLOCK

---

**Algorithm 1** Adversarially Robust Anomaly Detection through Spurious Negative Pair Mitigation

---

$\mathcal{T} \leftarrow$ {Color Jitter, Horizontal Flip, Grayscale, ...}        ▷ Set of k light augmentations
$T \leftarrow$ {Rotation, Elastic, Distortion, ...}        ▷ Set of k hard augmentations

**function** CLASSIFIER_GMM_TRAINER(training_data, $T$)
     $synthetic\_data \leftarrow \{(T_i(\text{training\_data}), i) \mid T_i \in T, i \in \text{range}(T)\}$      ▷ Create k-class cls dataset
     Train a k-class classifier $C$ on $synthetic\_data$
     $e_{train} \leftarrow C(\text{training\_data})$      ▷ Obtain embeddings from the classifier
     Fit GMM on $e_{train}$
     $\lambda \leftarrow 0.05$
     **return** $C$, GMM, $\lambda$
**end function**

**function** $\Upsilon(X_{normal}, T, C, \text{GMM}, \lambda)$
     **while** $p\_value > \lambda$ **do**
         $transforms\_seq \leftarrow$ Sample a random sequence of transforms from $T$
         $x_{p-anomaly} \leftarrow transforms\_seq(x_{normal})$    ▷ Apply hard transformations to create a pseudo-anomaly
         $e_{p-anomaly} \leftarrow C(x_{p-anomaly})$      ▷ Obtain embeddings from the classifier
         $p\_value \leftarrow$ GMM.P_Value $(e_{p-anomaly})$      ▷ Compute P-Value of embeddings
     **end while**
     **return** $x_{p-anomaly}$
**end function**

**function** PSEUDO_ANOMALY_GENERATOR($X_{normal}, T, C, \text{GMM}, \lambda$)
     $X_{p-anomaly} = \{\}$
     **for** $x_{normal}$ **in** $X_{normal}$ **do**      ▷ iterate over batch of normal data
         $x_{p-anomaly} \leftarrow \Upsilon(X_{normal}, T, C, \text{GMM}, \lambda)$
         $X_{p-anomaly}$.Add($x_{p-anomaly}$)
     **end for**
     **return** $X_{p-anomaly}$      ▷ Return the generated pseudo-anomaly sample
**end function**

**function** PGD(x, y, $\mathcal{F}, \mathcal{G}, \mathcal{H}$, pgd_steps, $\alpha, \epsilon$)
     $x\_adv \leftarrow x$
     **for** step **in** pgd_steps **do**
         grad = Compute_Gradient( $[\mathcal{L}_{\text{COBRA}}(\mathcal{F}, \mathcal{G}; x\_adv) + \mathcal{L}_{\text{CLS}}(\mathcal{F}, \mathcal{H}; x\_adv, y)], x\_adv$)
         ▷ Compute the gradient of the loss with respect to the input $x\_adv$
         $x\_adv = x\_adv + \alpha * sign(grad)$
         $x\_adv = clip(x\_adv, x - \epsilon, x + \epsilon)$      ▷ Gradient ascent and projection to valid $\epsilon$-ball
         $x\_adv = clip(x\_adv, 0, 1)$
     **end for**
     **return** $x\_adv$
**end function**

**function** ADVERSARIAL_TRAINING_COBRA(training_data, $\mathcal{F}, \mathcal{G}, \mathcal{H}, C, \text{GMM}, \lambda$, pgd_steps, $\alpha, \epsilon$)
     **for** $\mathcal{B}_{normal}$ **in** training_data **do**
         $\mathcal{B}_{p-anomaly} \leftarrow$ Pseudo_Anomaly_Generator($\mathcal{B}_{normal}, T, C, \text{GMM}, \lambda$)
         $\mathcal{B} \leftarrow$ Concatenate($\mathcal{B}_{normal}, \mathcal{B}_{p-anomaly}$)
         $Y \leftarrow [0] \times \|\mathcal{B}_{normal}\| + [1] \times \|\mathcal{B}_{p-anomaly}\|$
         $\tau_1, \tau_2 =$ Sample tow random recuence of transforms from $\mathcal{T}$
         $\mathcal{B}_{adv} \leftarrow \{\}$
         **for** x, y **in** $(\mathcal{B}, Y)$ **do**
             $x_1, x_2 \leftarrow \tau_1(x), \tau_2(x)$
             P($x_1$), P($x_2$) $\leftarrow \{x_2\}, \{x_1\}$
             N(x) $\leftarrow \{\tau_1(x') : x' \in \mathcal{B} \setminus \{x\}\} \cup \{\tau_2(x') : x' \in \mathcal{B} \setminus \{x\}\}$      ▷ N(x)=N($x_1$)=N($x_2$)
             $x\_adv \leftarrow$ PGD(x, y, $\mathcal{F}, \mathcal{G}, \mathcal{H}$, pgd_steps, $\alpha, \epsilon$)
             $\mathcal{B}_{adv}$.Add($x_{adv}$)
         **end for**
         $\mathcal{L} = 0$
         **for** $x, x_{adv}, y$ **in** $(\mathcal{B}, \mathcal{B}_{adv}, Y)$ **do**
             $x_1, x_2 \leftarrow \tau_1(x), \tau_2(x)$
             P($x_1$), P($x_2$), P($x_{adv}$) $\leftarrow \{x_2, x_{adv}\}, \{x_1, x_{adv}\}, \{x_1, x_2\}$
             N(x) $\leftarrow \{\tau_1(x') : x' \in \mathcal{B} \setminus \{x\}\} \cup \{\tau_2(x') : x' \in \mathcal{B} \setminus \{x\}\}$
             $\cup \{x'_{adv} : x'_{adv} \in \mathcal{B}_{adv} \setminus \{x_{adv}\}\}$
             $\mathcal{L}$ += $\mathcal{L}_{\text{COBRA}}(\mathcal{F}, \mathcal{G}; x) + \mathcal{L}_{\text{CLS}}(\mathcal{F}, \mathcal{H}; [x, x_{adv}], [y, y])$
         **end for**
         Update Networks($\mathcal{F}, \mathcal{G}, \mathcal{H}$) using $\mathcal{L}$
     **end for**
**end function**

**function** MAIN(epochs, training_data, $\mathcal{F}, \mathcal{G}, \mathcal{H}$, pgd_steps, $\alpha, \epsilon$)
     $C$, GMM, $\lambda \leftarrow$ Classifier_GMM_Trainer(training_data, $T$)
     **for** epoch **in** epochs **do**
         Adversarial_Training_COBRA(training_data, $\mathcal{F}, \mathcal{G}, \mathcal{H}, C, \text{GMM}, \lambda$, pgd_steps, $\alpha, \epsilon$)
     **end for**
**end function**

MAIN(epochs, training_data, $\mathcal{F}, \mathcal{G}, \mathcal{H}$, 10, $\alpha, \epsilon$)

---

## B    RELATED WORK

**Previous AD Methods.**  Recent standard AD methods can be categorized into two types: transfer learning based and CL based methods. Transfer learning-based methods utilize a trained model on a large dataset as a backbone and leverage its rich features for the AD task. This approach is evident in methods including PANDA Reiss et al. (2021), Transformaly Cohen and Avidan (2021), Patchcore Roth et al. (2021), and Fastflow Yu et al. (2021). On the other hand, CL framework has demonstrated its superiority by extracting discriminative features, as showcased by CSI Tack et al. (2020), MSAD Reiss and Hoshen (2021), Recontrast Guo et al. (2024), and Draem Zavrtanik et al. (2021). Extending transfer learning based methods to an adversarial setting is not feasible because pre-trained features, which act as a key, are not robust, necessitating a new training paradigm. This limitation inspired us to adopt CL for adversarial training, which also aligns with the unlabeled nature of AD. There have been a few efforts to propose robust AD methods, including PrincipaLS Lo et al. (2022), OCSDF Béthune et al. (2023), and APAE Goodge et al. (2021). However, their results on even tiny datasets are less than random detection. Details about each mentioned method can be found in Appendix C.

## C    DETAILS OF RELATED WORK

### C.1    PREVIOUS AD METHODS

There has been some efforts to develop a robust AD method. PrincipaLS employs a novel latent space manipulation technique to adjust the representations of data point with optimizing a robustness criterion designed to minimize the model's sensitivity to adversarial perturbations. OCSDF leverages the Signed Distance Function to delineate the boundary of a data distribution. Through the employment of 1-Lipschitz neural networks, it adeptly approximates normality scores, thus enhancing robustness to adversarial perturbations. APAE introduces the approximate projection autoencoder as a defense mechanism, integrating gradient descent on latent embeddings and feature-weighting normalization to enhance detection robustness. it worth noting we exclude robust OOD detection from our experiments, where their method have been develpoed by relying on labels and could not extend to AD setup. ZARND a introduces a robust method enhancing anomaly detection by integrating robust features from pretrained models with nearest-neighbor algorithms. The approach significantly improves robustness against adversarial attacks, which traditionally degrade ND performance. By leveraging features from adversarially robust models and employing k-Nearest Neighbors (k-NN) for anomaly scoring Mirzaei et al. (2022; 2024c); Salehi et al. (2021); Mirzaei and Mathis (2024); Mirzaei et al.; Moakhar et al. (2023); Mirzaei et al. (2024d); Jafari et al. (2024); Taghavi et al. (2023a); Rahimi et al. (2024a); Taghavi et al. (2023b;c); Taghavi and Mirzaei (2024); Ebrahimi et al. (2024a;b); Rahimi et al. (2024b).

### C.2    AUXILIARY ANOMALY SAMPLE CRAFTING

Previous studies on anomaly generation often struggle with producing samples that are either too similar to normal instances (distant anomalies) or inadvertently create samples that still belong to the normal category. Conversely, many studies have underscored the benefit of using related auxiliary anomaly samples to enhance detector performance. In light of this, we have proposed COBRA, which, unlike its counterparts such as Dream-OOD, does not require an extra dataset and performs well in an unsupervised setting where labels for normal samples are not available.

COBRA aligns with the distribution of the normal dataset, effectively generating informative pseudo-anomalies within the pixel space. It employs a thresholding technique to sift out inaccuracies, all while obviating the need for supplementary datasets.

## D    DISTRIBUTION AWARE HARD TRANSFORMATION

In this section, we elaborate on the practical implementation details of our proposed pseudo-anomaly crafting strategy. As discussed in the main text of the paper, we set the hyperparameter $\beta$ to $0.05$. To assess the robustness of our model with respect to this parameter, we conducted extensive experiments over a broader range, specifically $[0.02, 0.20]$. The stability of our model's performance across this range is demonstrated in the experimental results (refer to Table 7). These findings underscore

the robustness of our approach in handling varying hyperparameter settings without significant performance degradation.

Table 7: Performance of the model for different values of the hyperparameter $\beta$ used in pseudo-anomaly crafting. Results are shown in each table cell as 'Clean/PGD-1000'. (Evaluations are done on CIFAR-10 and CIFAR-100 datasets under a PGD-1000 attack with $\epsilon = \frac{4}{255}$, and on other high-resolution datasets with $\epsilon = \frac{2}{255}$).

| $\beta$ | Datasets | | | | | | |
|---|---|---|---|---|---|---|---|
| | cifar10 | cifar100 | MVTecAD | CityScapes | VisA | ISIC2018 | DAGM |
| 0.02 | 82.7 / 61.8 | 75.4 / 52.3 | 88.0 / 74.1 | 79.9 / 57.5 | 73.7 / 70.1 | 82.3 / 55.6 | 80.4 / 55.1 |
| 0.05 | 83.7 / 62.3 | 76.9 / 51.7 | 89.1 / 75.1 | 81.7 / 56.2 | 75.2 / 73.8 | 81.3 / 56.1 | 82.4 / 56.8 |
| 0.08 | 82.3 / 62.1 | 74.2 / 50.8 | 87.3 / 73.9 | 81.3 / 55.9 | 76.1 / 73.0 | 81.0 / 57.0 | 82.1 / 55.7 |
| 0.11 | 81.9 / 61.3 | 74.8 / 51.0 | 88.1 / 74.4 | 80.8 / 55.1 | 74.2 / 72.7 | 80.6 / 55.2 | 81.3 / 57.0 |
| 0.14 | 80.8 / 60.9 | 77.2 / 51.8 | 89.0 / 74.5 | 80.6 / 56.2 | 75.8 / 72.6 | 79.8 / 56.4 | 79.5 / 54.6 |
| 0.17 | 79.5 / 59.3 | 75.8 / 50.4 | 86.9 / 75.0 | 79.6 / 56.7 | 74.2 / 72.9 | 79.3 / 54.1 | 79.0 / 53.7 |
| 0.20 | 80.1 / 58.9 | 76.3 / 49.9 | 89.2 / 74.2 | 80.0 / 54.2 | 73.9 / 72.1 | 80.3 / 55.3 | 81.9 / 54.7 |

## D.1 HARD TRANSFORMATION

We employed a series of hard transformations to create the $T$ set. The transformations and their respective hyperparameters are as follows:

- **Jigsaw**: Images were divided into a 2x2 grid, and the tiles were randomly permuted. This transformation disrupts the spatial continuity of image features.

- **Random Erasing**: A random rectangular region in the image was erased, with a size proportional to the image area and an aspect ratio randomly chosen between 10% and 50%.

- **CutPaste**: A square region, with side length varying from 10% to 50% of the image width, was cut and pasted into a different location within the same image.

- **Rotation**: Images were rotated by a random angle within $\pm 90$ degrees to introduce a moderate level of distortion.

- **Extreme Blurring**: Applied a Gaussian blur with a kernel size of up to 5% of the image width and a high variance ($\sigma = 2.5$), resulting in significant blurring.

- **Intense Random Cropping**: Random sections, sized between 50% and 80% of the original image size, were cropped to challenge the model with incomplete patterns.

- **Noise Injection**: Gaussian noise with a mean of 0 and a standard deviation of 0.1 was added to the images.

- **Extreme Cropping**: Cropped the image to retain only 40% to 60% of its original size, focusing on the center to ensure a significant deviation from the original distribution.

- **Mixup**: To create composite images that merge features from both source images in a more challenging manner, we combined pairs of images using a blending coefficient $\alpha$, drawn from a Beta distribution Beta$(\alpha, \alpha)$ with $\alpha$ set to 0.1. This lower value of $\alpha$ results in a higher variance of the mixing coefficients, producing images that are significantly more blended than with a higher $\alpha$ value. This aggressive mixing ensures the crafting of pseudo-anomaly samples.

- **Cutout**: Square regions with a side length of 25% of the image width were filled with a constant value to simulate occlusion.

- **CutMix**: Portions from one image were cut and pasted onto another, with the cut region's size approximately 20% of the image area.

**Quality of Generated Abnormal Data.** In Table 5 of our paper, we evaluated our crafting strategy by comparing it with alternative methods. Additionally, Figure 2 presents samples that visually compare our results with others. To address any concerns further, we employed the FID metric, which measures the distance from a normal distribution, as well as the DC metric Naeem et al. (2020) to assess diversity. A lower FID indicates that the crafted anomalies are more similar to normal

instances, while a higher DC suggests greater diversity among the samples. Table 8 compares the quality of anomalies generated by our method with those from other alternatives, using the FID and DC metrics.

Table 8: Assessment of Generated Anomaly Data Quality Compared to Alternative Methods

| Dataset | Metric | GOE | FITYMI | Dream-OOD | Dream-OOD | Ours |
|---------|--------|-----|--------|-----------|-----------|------|
| MVTecAD | FID ↓ | 245 | 163 | 227 | 227 | 129 |
|         | DC ↑ | 0.53 | 0.29 | 0.75 | 0.75 | 0.84 |
| ImageNet | FID ↓ | 118 | 145 | 98 | 98 | 107 |
|          | DC ↑ | 0.64 | 0.38 | 0.87 | 0.87 | 0.58 |
| CIFAR10 | FID ↓ | 76 | 86 | 72 | 72 | 54 |
|         | DC ↑ | 0.45 | 0.36 | 0.56 | 0.56 | 0.72 |
| FMNIST | FID ↓ | 42 | 37 | 64 | 64 | 32 |
|        | DC ↑ | 0.31 | 0.17 | 0.45 | 0.45 | 0.63 |
| **Average** | FID ↓ | 120.3 | 107.8 | 115.3 | 115.3 | **80.3** |
|             | DC ↑ | 0.48 | 0.3 | 0.65 | 0.65 | **0.69** |

## D.2 POSITIVE TRANSFORMATION

Consistent with the self-supervised learning literature Chen et al. (2020b); He et al. (2020), we employ mild transformations that generate samples with minimal visual differences while fully preserving semantics. These include *color jitter, random grayscale conversion, and random cropping (cropping the image to retain 80% to 100% of its original size)*. These transformations are designed to subtly alter the appearance of images without changing their underlying content, thereby enabling the model to learn robust features that are invariant to minor perturbations.

## E ADDITIONAL ABLATION STUDY

Here, we conduct further experiments to thoroughly evaluate COBRA across diverse settings, highlighting its key components and their substantial impact.

**Clean Training** In this scenario, we skipped adversarial training and instead trained COBRA with standard training, keeping all other components fixed. The results indicate that clean performance increased from **84.1** to **90.7**, demonstrating COBRA's superiority in various scenarios of training and evaluating. The results are presented in Table 9.

**Anomaly Score** Instead of utilizing our proposed Anomaly Score, we explored alternative approaches while keeping other components unchanged. Specifically, our default anomaly score is based on the similarity between a test sample and normal training samples. We substituted this anomaly score with logits provided by our binary classifier head. Specifically, the COBRA results reported in the main paper are based on the $A(x)$ anomaly score, which is defined as:

$$A(x) = - \max_{x^i \in D_{\text{train}}} \left\{ \langle \mathcal{G}(\mathcal{F}(x)), \mathcal{G}(\mathcal{F}(x^i)) \rangle \right\},$$

Instead, we replaced that with $A'(x) = p(\mathcal{H}(\mathcal{F}(x))|y = 1)$, which denotes the probability of belonging to the pseudo-anomaly class as assigned by the binary classifier head. Additionally, we considered $A(x) + A'(x)$ as another alternative. As the results presented in Table 10 demonstrate, all strategies achieve significant performance with minor differences.

**Ablation Study on Distribution-Aware Hard Transformation**
In this section, we utilize the feature extractor $C$ as an anomaly detector to demonstrate that its

Table 9: Performance of COBRA trained with standard training (without adversarial training) across various datasets.

| Evaluation setting | Datasets | | | | | | | | | |
|---|---|---|---|---|---|---|---|---|---|---|
| | CIFAR10 | CIFAR100 | MNIST | FMnist | SVHN | ImagenNet30 | MVTechAD | VisA | DAGM | ISIC2018 |
| Clean | 94.3 | 93.7 | 95.1 | 93.4 | 96.0 | 92.7 | 95.2 | 78.1 | 90.8 | 83.4 |
| Adversarial | 1.7 | 4.6 | 1.0 | 3.8 | 2.5 | 4.9 | 5.8 | 2.3 | 4.8 | 4.2 |

Table 10: Ablation study on different score functions. Results are AUROC (%)

| Dataset | Anomaly Score | | |
|---|---|---|---|
| | $A(x) + A'(x)$ | $A'(x)$ | $A(x)$ (default) |
| MVTecAD | 88.4 / 74.6 | 83.1 / 72.8 | 89.1 / 75.1 |
| CIFAR10 | 81.9 / 60.2 | 76.1 / 55.7 | 83.7 / 62.3 |
| FMNIST | 89.6 / 87.4 | 88.5 / 84.8 | 93.1 / 89.6 |
| CIFAR100 | 72.6 / 51.2 | 70.4 / 48.3 | 76.9 / 51.7 |

Table 11: $C$'s performance in the AD task across various datasets.

| Methods | Datasets | | | | | | | | | |
|---|---|---|---|---|---|---|---|---|---|---|
| | CIFAR10 | CIFAR100 | MNIST | FMnist | SVHN | ImagenNet30 | MVTechAD | VisA | DAGM | ISIC2018 |
| COBRA | 77.1 | 72.4 | 86.0 | 83.6 | 74.7 | 80.3 | 75.2 | 64.9 | 78.5 | 73.2 |

learned features from the normal training set are meaningful and significantly surpass random detection, highlighting its role in crafting effective pseudo-anomaly samples. Results presented in Table 11.

**Ablation Study on Thresholding**
Here, while keeping all components of COBRA constant, we skip the thresholding strategy and instead use a random subset of hard transformations for crafting pseudo-anomaly samples. The results, indicated in Table 12, suggest that this approach leads to decreased performance due to the failure to filter incorrect pseudo-anomaly samples (those that still belong to the normal set).

Table 12: Impact of Skipping Thresholding on Pseudo-Anomaly Generation in COBRA

| Methods | Datasets | | | | | | | | | |
|---|---|---|---|---|---|---|---|---|---|---|
| | CIFAR10 | CIFAR100 | MNIST | FMnist | SVHN | ImagenNet30 | MVTechAD | VisA | DAGM | ISIC2018 |
| COBRA | 78.6 / 58.1 | 65.4 / 46.5 | 94.3 / 91.5 | 90.1 / 86.5 | 84.8 / 53.2 | 72.5 / 55.7 | 76.0 / 68.2 | 78.4 / 47.9 | 74.6 / 52.0 | 73.8 / 51.4 |

# F  ANALYZING COBRA'S STABILITY AND EFFECTIVENESS

Figure 4 represents COBRA's loss values and its detection performance at each epoch of training for both clean data and data subjected to PGD attack, demonstrating the stability of COBRA's loss function. The experiment was conducted in a one-class setup using the MVETEC-AD and FMNIST datasets. Note that the loss values have been normalized between 0 and 1, and the loss values for the PGD data are higher than those for the clean data. Despite this, the figure underscores the stability of our loss function, which remains consistent across different training conditions.

# G  $A^3$ TO ALL METHODS

Here, we present more detailed results of previous anomaly detection (AD) works and their performance under $A^3$ (See Table 13). Their respective performance against PGD-1000 attacks has been provided in the main paper.

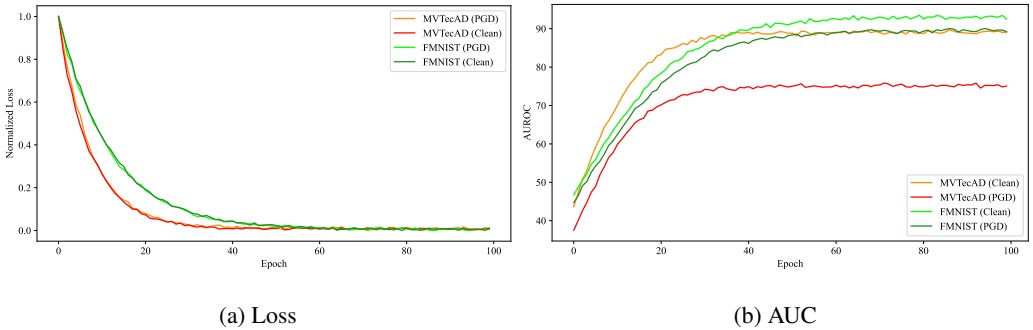

(a) Loss                                (b) AUC

Figure 4: This figure depicts COBRA's loss values and detection performance for each epoch of training on both clean data and data under PGD attack, demonstrating the stability of COBRA's loss function. The experiment was carried out in a one-class setup using the MVETEC-AD and FMNIST datasets.

Table 13: AUROC (%) of various methods for One Class AD methods under AutoAttack.

*These models are trained to be adversarially robust.

| Dataset | Method | | | | | | | | | |
|---|---|---|---|---|---|---|---|---|---|---|
| | DeepSVDD | CSI | DN2 | PANDA | MSAD | Transformaly | PatchCore | PrincipaLS* | OCSDF* | APAE* |
| CIFAR10 | 16.6 | 3.3 | 2.5 | 1.2 | 0.7 | 2.4 | 2.5 | 29.6 | 26.9 | 2.0 |
| CIFAR100 | 14.3 | 1.2 | 0.7 | 1.1 | 10.7 | 7.3 | 3.5 | 24.6 | 19.8 | 0.9 |
| MNIST | 15.4 | 1.7 | 1.0 | 0.7 | 14.1 | 11.6 | 2.4 | 77.3 | 66.2 | 28.6 |
| Fashion-MNIST | 48.1 | 8.5 | 2.2 | 9.7 | 3.7 | 2.8 | 2.8 | 67.5 | 59.6 | 12.3 |
| SVHN | 4.3 | 0.8 | 0.2 | 0.4 | 0.1 | 0.2 | 1.1 | 8.9 | 6.1 | 0.3 |
| ImageNet30 | 12.4 | 2.7 | 1.2 | 0.5 | 1.4 | 1.8 | 2.3 | 23.8 | 20.6 | 1.1 |

## H    EVALUATING OUR MODEL UNDER VARIOUS ATTACKS WITH DIVERSE EPSILON VALUES

In this section, we evaluate our model's performance under adversarial attacks, specifically focusing on $\ell_\infty$ and $\ell_2$ PGD attacks across various epsilon values. Table 14 presents the results for the $\ell_\infty$ PGD-1000 attacks, indicating that the model maintains strong accuracy for lower epsilon values, with some decline in performance observed as epsilon increases. Similarly, Table 15 illustrates the model's robustness under $\ell_2$ PGD attacks, demonstrating consistent accuracy across lower epsilon values, with a gradual trend observed at higher epsilon values. These results highlight the model's resilience to adversarial attacks while also indicating areas for potential improvement.

Table 14: Evaluation of our model under various $\ell_\infty$ PGD-1000 attacks across different $\epsilon$ values. Although the model was trained using $\ell_\infty$ PGD-10 with $\epsilon = 2/255$ for high-resolution images and $\epsilon = 4/255$ for low-resolution images, the evaluation settings were modified to test its robustness against stronger attacks.

| Epsilon | MVTec | VisA | ImageNet | CityScapes | ISIC2018 | CIFAR10 | FMNIST | Avg. |
|---|---|---|---|---|---|---|---|---|
| 0 | 89.1 | 75.2 | 85.2 | 81.7 | 81.3 | 83.7 | 93.1 | 84.2 |
| 1/255 | 81.6 | 74.6 | 68.1 | 65.7 | 62.8 | 77.0 | 91.1 | 74.4 |
| 2/255 | 75.1 | 73.8 | 57.0 | 56.2 | 56.1 | 72.9 | 90.2 | 68.7 |
| 3/255 | 64.7 | 67.2 | 54.1 | 50.9 | 51.8 | 69.3 | 89.9 | 64.0 |
| 4/255 | 56.1 | 52.9 | 45.2 | 47.1 | 49.3 | 62.3 | 89.6 | 57.5 |
| 5/255 | 49.5 | 46.3 | 43.7 | 40.9 | 45.5 | 60.8 | 89.2 | 53.7 |
| 6/255 | 42.8 | 40.7 | 39.4 | 38.6 | 42.1 | 56.3 | 89.2 | 49.9 |
| 7/255 | 38.1 | 33.9 | 34.5 | 35.2 | 39.6 | 51.8 | 89.1 | 46.0 |
| 8/255 | 34.2 | 30.7 | 28.6 | 32.8 | 35.5 | 47.4 | 89.1 | 42.6 |

Table 15: Evaluation of our model under various $\ell_2$ PGD attacks across different $\epsilon$ values. Although the model was trained using $\ell_\infty$ PGD-10 with $\epsilon = 2/255$ for high-resolution images and $\epsilon = 4/255$ for low-resolution images, the evaluation settings were modified to test its robustness under stronger $\ell_2$-norm adversarial attacks.

| Epsilon | MVTec | VisA | ImageNet | CityScapes | ISIC2018 | CIFAR10 | FMNIST | Avg. |
|---------|-------|------|----------|------------|----------|---------|--------|------|
| 0 | 89.1 | 75.2 | 85.2 | 81.7 | 81.3 | 83.7 | 93.1 | 84.2 |
| 16/255 | 87.1 | 74.9 | 77.8 | 77.9 | 76.8 | 73.8 | 91.0 | 79.9 |
| 32/255 | 85.4 | 74.8 | 75.1 | 77.8 | 74.2 | 64.1 | 89.9 | 77.3 |
| 64/255 | 83.8 | 74.2 | 71.0 | 72.6 | 71.3 | 55.4 | 88.9 | 73.9 |
| 128/256 | 79.6 | 73.1 | 65.0 | 66.7 | 60.5 | 48.9 | 87.6 | 68.8 |

# I    IMPLEMENTATION DETAILS

We employ ResNet-18 as the foundational encoder network ($f_\theta$), accompanied by an auxiliary head ($g_\phi$) consisting of a 2-layer multi-layer perceptron with a 128-dimensional embedding. For optimization, COBRA is trained for 100 epochs using the LARS optimizer, with a weight decay of $1 \times 10^{-6}$ and a momentum of 0.9. To schedule the learning rate, we adopt a linear warmup for the initial 10 epochs, gradually increasing the learning rate to 1.0. Subsequently, we use a cosine decay schedule without restarts. The batch size for COBRA is set to 128. Our experiments were conducted using NVIDIA GeForce RTX 3090 GPUs (24GB).

**Training Computational Cost.** COBRA comprises two main steps: (i) generating pseudo-anomaly samples from the normal training set, and (ii) adversarially training a model using both normal and crafted pseudo-anomalies. In this section, we analyze the complexity of these steps. Figure 5 illustrates that COBRA achieves significant performance with low complexity in terms of time. Please note that the training time shown in the figure is calculated for a single class of the dataset. In Table 16, we compare the time efficiency of our method, both with and without adversarial training, against other state-of-the-art methods across multiple datasets.

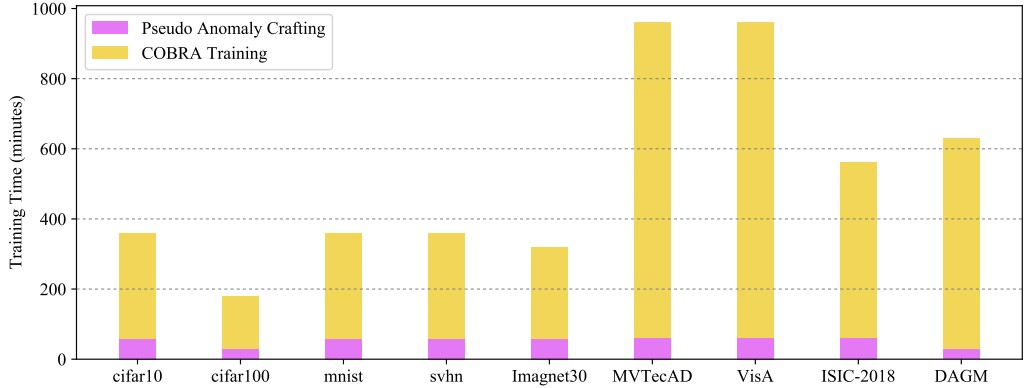

Figure 5: Computation cost

Table 16: Training Time Comparison (in hours) Across Different Methods and Datasets

| Method | MVTecAD | VisA | CityScape | ImageNet |
|--------|---------|------|-----------|----------|
| Transformaly | 30h | 70h | 6h | 200h |
| ReContrast | 25h | 60h | 5h | 180h |
| Ours (with adv. training) | 95h | 250h | 20h | 500h |
| Ours (without adv. training) | 9h | 23h | 2h | 70h |

**Evaluation of Computational Cost.** After training, we freeze our model and extract features from the training samples to create an embedding bank, which maps images from high-dimensional spaces (e.g., (1000, 3, 224, 224)) to a lower-dimensional space (e.g., (1000, 256)). During inference, for each test sample, we compute its features using the frozen model and compare its similarity to the precomputed embedding bank. This process is computationally efficient since it operates with a frozen model and in a low-dimensional space. To the best of our knowledge, using similarity-based methods, such as k-NN in embedding spaces for detection tasks, is common and well-established in the literature (e.g., MSAD). Furthermore, we provide detailed computational time information, excluding the time required for feature bank creation as it can be precomputed. The anomaly score computation times on a 3090 GPU, in comparison to other methods, are presented in Table 17, with results averaged over 100,000 inferences. For our model, embedding extraction takes 1-3 ms, matrix multiplication 1-2 ms, and finding the maximum value takes less than 1 ms.

Table 17: Per-Image Evaluation Time Comparison (in milliseconds) Across Different Methods

| Metric | CSI | MSAD | Transformaly | ReContrast | OCSDF | COBRA (Ours) |
|---|---|---|---|---|---|---|
| **Per-Image Eval Time** | 4 ms | 4 ms | 5 ms | 5 ms | 3 ms | 4 ms |

## J  PER-CLASS RESULTS

Anomaly detection evaluation scenarios can be categorized into one-class anomaly detection and unlabeled multi-class setups, as mentioned in the experiments section. In this section, we provide details of the one-class classification setup for COBRA on the reported dataset, as presented in Tables 18, 19.

.

Table 18: The detailed AUROC scores of the class-wise experiments for One-Class Anomaly Detection setting with PGD-1000 $\epsilon = \frac{4}{255}$ in CIFAR10, CIFAR100, MNIST, Fashion-MNIST, SVHN datasets.

(a) MNIST

| Method | Attack | Class | | | | | | | | | | Average |
|---|---|---|---|---|---|---|---|---|---|---|---|---|
| | | 0 | 1 | 2 | 3 | 4 | 5 | 6 | 7 | 8 | 9 | |
| Ours | Clean | 97.6 | 39.9 | 99.6 | 99.2 | 98.9 | 98.3 | 99.6 | 95.6 | 99.7 | 99.6 | 92.8 |
| | BlackBox | 92.4 | 99.6 | 98.9 | 95.1 | 97.5 | 94.6 | 99.1 | 96.9 | 96 | 97.2 | 96.8 |
| | PGD-100 | 92.4 | 99.3 | 98.3 | 95.1 | 96.9 | 94.4 | 98.7 | 96.7 | 96 | 96.6 | 96.4 |
| | $A^3$ | 91.7 | 98.3 | 97.8 | 94.3 | 94.5 | 94 | 97.3 | 94.3 | 95.1 | 94.4 | 95.2 |

(b) Fashion-MNIST

| Method | Attack | Class | | | | | | | | | | Average |
|---|---|---|---|---|---|---|---|---|---|---|---|---|
| | | 0 | 1 | 2 | 3 | 4 | 5 | 6 | 7 | 8 | 9 | |
| COBRA | Clean | 89.5 | 99.6 | 92.7 | 91.4 | 86.7 | 97.1 | 82.5 | 97 | 96.2 | 98.4 | 93.1 |
| | BlackBox | 88.7 | 99.5 | 85.8 | 85.7 | 83.3 | 95.3 | 82.6 | 97.9 | 92.6 | 97.8 | 90.9 |
| | PGD-100 | 85.4 | 99 | 85.1 | 84.6 | 83 | 95.3 | 80.9 | 95.1 | 90.7 | 97.4 | 89.6 |
| | $A^3$ | 83.7 | 96.5 | 85 | 83.9 | 81.7 | 93.5 | 77.2 | 91.5 | 87 | 93.6 | 87.4 |

(c) CIFAR10

| Method | Attack | Class | | | | | | | | | | Average |
|---|---|---|---|---|---|---|---|---|---|---|---|---|
| | | 0 | 1 | 2 | 3 | 4 | 5 | 6 | 7 | 8 | 9 | |
| Ours | Clean | 79.4 | 96.3 | 75.4 | 71.3 | 76 | 84.3 | 76.1 | 94.5 | 92.8 | 91.1 | 83.7 |
| | BlackBox | 75.9 | 96.1 | 75.2 | 68.4 | 73.4 | 84 | 74.2 | 92 | 89.9 | 89.8 | 81.8 |
| | PGD-100 | 64.7 | 80.4 | 47.9 | 39.9 | 49.5 | 60.7 | 52.9 | 74.8 | 80.3 | 71.7 | 62.3 |
| | $A^3$ | 64.5 | 77.7 | 46.6 | 39.8 | 48.8 | 58.3 | 50.1 | 72.9 | 80.3 | 68.7 | 60.7 |

(d) SVHN

| Method | Attack | Class | | | | | | | | | | Average |
|---|---|---|---|---|---|---|---|---|---|---|---|---|
| | | 0 | 1 | 2 | 3 | 4 | 5 | 6 | 7 | 8 | 9 | |
| Ours | Clean | 793.7 | 89.6 | 88.8 | 79.3 | 88.3 | 88.9 | 89.8 | 93.1 | 90.4 | 90.9 | 89.3 |
| | BlackBox | 88.8 | 88.9 | 85.7 | 75.2 | 83.7 | 84.1 | 86.9 | 90.5 | 89.8 | 89.9 | 86.4 |
| | PGD-100 | 61.2 | 61.8 | 54.5 | 45.4 | 59.4 | 55.2 | 61 | 67.7 | 54.4 | 61.4 | 58.2 |
| | $A^3$ | 60.1 | 61.6 | 52.4 | 43.6 | 56.6 | 55.6 | 59.7 | 65.3 | 52.4 | 58.7 | 56.7 |

(e) CIFAR100

| Method | Attack | Class | | | | | | | | | | | | | | | | | | | | Average |
|---|---|---|---|---|---|---|---|---|---|---|---|---|---|---|---|---|---|---|---|---|---|---|
| | | 0 | 1 | 2 | 3 | 4 | 5 | 6 | 7 | 8 | 9 | 10 | 11 | 12 | 13 | 14 | 15 | 16 | 17 | 18 | 19 | |
| Ours | Clean | 68.9 | 73.1 | 79.4 | 71.9 | 84.24 | 70.1 | 79.5 | 72.3 | 71.7 | 85.4 | 83.34 | 78.8 | 80 | 57.5 | 80.4 | 69.6 | 62.5 | 92.4 | 88.9 | 88.5 | 76.9 |
| | BlackBox | 65.6 | 72.7 | 79 | 70.2 | 80.7 | 68.2 | 75.4 | 72.1 | 71.3 | 84.1 | 79.7 | 76.5 | 78.6 | 54.4 | 76.7 | 65.4 | 60.7 | 91.3 | 85 | 87.4 | 74.6 |
| | PGD-100 | 43.6 | 51.2 | 47.8 | 51.2 | 57.2 | 44.5 | 55.3 | 38.5 | 40.2 | 70.4 | 72.4 | 50.2 | 48.7 | 29.6 | 50.7 | 39.9 | 36 | 75.5 | 63.5 | 69.3 | 51.7 |
| | $A^3$ | 42.6 | 48.1 | 47.4 | 51.2 | 56.6 | 42.9 | 52.9 | 35.8 | 40.1 | 69.5 | 72.4 | 48.3 | 47.3 | 27.3 | 49.3 | 37.4 | 32.4 | 72.5 | 62.3 | 66.7 | 60.7 |

Table 19: The detailed AUROC scores of the class-specific experiments for One-Class Anomaly Detection setting with PGD-1000 $\epsilon = \frac{4}{255}$ in Imagenet30 dataset.

(a) ImageNet30

| Method | Attack | Class | | | | | | | | | | Average |
|--------|--------|------|------|------|------|------|------|------|------|------|------|---------|
| | | 0 | 1 | 2 | 3 | 4 | 5 | 6 | 7 | 8 | 9 | |
| Ours | Clean | 74.9 | 98.1 | 99.7 | 73.5 | 83.4 | 96.7 | 93.2 | 83.3 | 82.7 | 74 | 85.2 |
| | BlackBox | 72.3 | 93.3 | 95.6 | 71.1 | 80.1 | 96 | 89.1 | 82.2 | 78.8 | 70.3 | 82.4 |
| | PGD-100 | 35.8 | 90.9 | 96.7 | 46.4 | 36.3 | 86.3 | 60.9 | 55.7 | 38.3 | 24.4 | 57 |
| | $A^3$ | 34.9 | 89.8 | 94.6 | 44.2 | 32.7 | 82.6 | 56.4 | 52.6 | 35.8 | 22.6 | 54.7 |

(b) ImageNet30

| Method | Attack | Class | | | | | | | | | | Average |
|--------|--------|------|------|------|------|------|------|------|------|------|------|---------|
| | | 10 | 11 | 12 | 13 | 14 | 15 | 16 | 17 | 18 | 19 | |
| Ours | Clean | 96.9 | 90.1 | 93.3 | 82.1 | 94.6 | 62.8 | 98.2 | 58.5 | 89.4 | 54.7 | 85.2 |
| | BlackBox | 96 | 87.1 | 91.8 | 78.3 | 90.3 | 62.5 | 95.4 | 55.5 | 88.7 | 51.9 | 82.4 |
| | PGD-100 | 71.3 | 62.8 | 75.5 | 36.6 | 76.6 | 16.4 | 88.1 | 17 | 58.5 | 24 | 57 |
| | $A^3$ | 67.1 | 61.6 | 70.9 | 35.7 | 72.6 | 13.8 | 87.1 | 15.9 | 55.7 | 22.2 | 54.7 |

(c) ImageNet30

| Method | Attack | Class | | | | | | | | | | Average |
|--------|--------|------|------|------|------|------|------|------|------|------|------|---------|
| | | 20 | 21 | 22 | 23 | 24 | 25 | 26 | 27 | 28 | 29 | |
| Ours | Clean | 95 | 88.1 | 97.2 | 96.5 | 80.2 | 69.2 | 83.5 | 92.9 | 78.3 | 93.7 | 85.2 |
| | BlackBox | 92.3 | 83.9 | 91.5 | 96.3 | 75.1 | 65.3 | 82.4 | 89.9 | 76.7 | 90.8 | 82.4 |
| | PGD-100 | 77.9 | 44.8 | 90.8 | 79.1 | 39.2 | 40.6 | 32.2 | 78.1 | 41.3 | 86.5 | 57 |
| | $A^3$ | 74.7 | 44.9 | 89.8 | 75.2 | 35.7 | 38.4 | 31.8 | 77.3 | 38.8 | 84.4 | 54.7 |

# K    ADVERSARIAL ATTACKS ADAPTATION

We evaluated COBRA's robustness against a variety of powerful attacks, including BlackBox, FGSM, CAA, AutoAttack, $A^3$, and PGD-1000. These attacks, originally designed to compromise classification tasks by exploiting the cross-entropy loss, were adapted for anomaly detection (AD) tasks, focusing on the anomaly scores of detector models. The aim was to generate perturbations that increase the anomaly score for normal test samples and decrease it for anomalous ones. As discussed in the preliminaries section, adapting AutoAttack (AA) Croce and Hein (2020) for AD tasks was particularly challenging. AutoAttack is an ensemble of different attack methods, such as FAB, multi-targeted FAB, Square Attack, APGDT, APGD with cross-entropy loss, and APGD with DLR loss. The main challenge in adaptation arises because attacks based on DLR loss assume the model's output includes at least three elements, an assumption valid for classification tasks on datasets with three or more classes but not applicable to AD tasks. Consequently, we replaced the DLR loss component in AutoAttack with a PGD attack. However, for the other attacks under consideration, no adjustments were necessary.

# L    DATASETS DETAILS

The MVTecAD is an industrial defect detection dataset used to evaluate AD methods. It consists of 4,096 normal and 1,258 anomaly samples, encompassing various types of texture defects. MVTecAD is under the CC-BY-NC-SA 4.0 license. VisA is another challenging dataset for industrial defect detection, comprising 9,621 normal and 1,200 anomaly samples. VisA is under the CC-BY 4.0 license. DAGM is a synthetic dataset created for defect detection on textured surfaces. To broaden the scope beyond traditional industrial scenarios, the Cityscapes dataset offers stereo videos from 50 cities, each meticulously annotated for 30 classes such as roads and buildings. We leverage this dataset by extracting 256x256 patches from its images to construct an anomaly detection dataset, focusing on the presence of anomaly objects within these patches. Anomaly classes encompass motorcycles, persons, riders, traffic signs, traffic lights, and bicycles, while other classes from Cityscapes are considered normal. Their code is released under the MIT license. Additionally, ISIC2018 is a skin disease dataset, available as task 3 of the ISIC2018 challenge. It contains seven classes. NV (nevus) is taken as the normal class, and the rest of the classes are taken as anomalies, following. The training set contains 6,705 normal images. The ISIC dataset is available under CC-BY-NC license. Furthermore, we utilize ImageNet30 Hendrycks et al. (2019a), an anomaly detection benchmark that selects 30 classes from ImageNet and employs a one-versus-rest setup for anomaly detection. This dataset is freely available to researchers for non-commercial use.

# M    DETAILED RESULTS

The figure presents the standard deviation and mean of COBRA (a specific algorithm or method) calculated over 5 separate runs for each experiment. This comprehensive data collection and analysis underscore the reliability and consistency of our experimental results. By reporting both the mean and standard deviation, we provide a clear depiction of the average performance and the variability, ensuring that the performance of COBRA is not only robust but also consistently reproducible across multiple trials.

Table 20: The standard deviation and mean of COBRA across 5 runs of each experiment are reported, demonstrating the consistency of our results.

| Statistics | Eval Type | Datasets | | | | | | | | | |
|---|---|---|---|---|---|---|---|---|---|---|---|
| | | CIFAR10 | CIFAR100 | MNIST | FMNIST | SVHN | ImageNet | VisA | CityScapes | DAGM | ISIC2018 |
| Mean ± STD | Clean | $83.7 \pm 0.52$ | $76.9 \pm 0.81$ | $92.8 \pm 0.47$ | $93.1 \pm 0.56$ | $89.3 \pm 0.39$ | $85.2 \pm 1.21$ | $75.2 \pm 1.03$ | $81.7 \pm 0.97$ | $82.4 \pm 0.87$ | $81.3 \pm 0.74$ |
| | Adv | $62.3 \pm 0.73$ | $51.7 \pm 0.91$ | $96.4 \pm 0.86$ | $89.6 \pm 0.91$ | $58.2 \pm 0.62$ | $57.0 \pm 1.41$ | $73.8 \pm 1.17$ | $56.2 \pm 1.31$ | $56.8 \pm$ | $56.1 \pm 1.42$ |

# N    SUPPLEMENTARY METRICS FOR COBRA ASSESSMENT

We found the Area Under the Receiver Operating Characteristic curve (AUROC) to be the most widely accepted and utilized metric. To further our exploration, we have provided additional results

using two supplementary metrics—AUPR and FPR95%—which have been utilized in some previous works Hendrycks et al. (2019b). In the table 21, we compare COBRA against TRANSFORMALY and ZARND, a recent detection method, using these metrics. FPR95% represents the false positive rate when 95% of the outliers are correctly detected; a lower FPR95% signifies better performance. Both AUROC and AUPR summarize a detection method's performance over various thresholds. Specifically, AUROC indicates the likelihood that an outlier is ranked higher in anomaly score compared to an in-distribution sample. Hence, higher AUROC and AUPR values denote superior performance, with an uninformative detector scoring an AUROC of 50%. To address the reviewer's concerns, we will consider including AUPR and FPR95% along with AUROC in our final manuscript.

Table 21: Methods and Metrics Comparison Across Different Datasets

| Methods | Metric | MVTec | VisA | ImageNet | CityScapes | ISIC2018 | CIFAR10 | FMNIST |
|---|---|---|---|---|---|---|---|---|
| Transformaly | AUROC ↑ | 88.5/2.2 | 85.5/0.0 | 99.0/2.9 | 87.4/4.5 | 86.6/3.9 | 98.3/3.7 | 94.4/7.4 |
| | AUPR ↑ | 85.6/3.9 | 83.7/4.1 | 94.1/2.3 | 86.0/2.6 | 83.5/2.8 | 99.6/1.4 | 98.2/0.0 |
| | FPR95% ↓ | 41.6/99.7 | 26.7/98.9 | 2.6/99.1 | 35.7/97.9 | 28.4/98.0 | 8.4/99.6 | 9.5/95.4 |
| ZARND | AUROC ↑ | 71.6/30.1 | 71.8/24.9 | 96.4/27.4 | 75.9/28.6 | 70.2/14.6 | 89.7/56.0 | 95.0/82.3 |
| | AUPR ↑ | 73.5/28.7 | 69.8/20.1 | 90.1/29.8 | 72.1/26.5 | 73.6/16.4 | 85.5/52.3 | 92.5/80.0 |
| | FPR95% ↓ | 41.2/69.2 | 44.1/73.9 | 7.3/64.8 | 40.1/66.4 | 26.8/75.3 | 26.4/55.9 | 9.6/23.3 |
| COBRA (Ours) | AUROC ↑ | 89.1/75.1 | 75.2/73.8 | 85.2/57.0 | 81.7/56.2 | 81.3/56.1 | 83.7/62.3 | 93.1/89.6 |
| | AUPR ↑ | 91.7/71.9 | 78.6/70.3 | 88.9/61.2 | 85.6/62.1 | 87.6/59.8 | 84.7/65.7 | 95.1/89.8 |
| | FPR95% ↓ | 18.5/36.8 | 35.8/38.7 | 24.9/54.7 | 29.3/55.0 | 30.1/52.7 | 26.4/43.9 | 6.7/17.3 |

## O  SOCIAL IMPACTS

Robust anomaly detection is crucial across many safety-critical domains like security, healthcare, finance, and manufacturing to identify potential threats, diseases, fraud or system faults before they cause harm. However, existing machine learning-based anomaly detectors are vulnerable to adversarial attacks that can make them miss anomalies or falsely flag normal data. Our work on COBRA presents a significant step towards developing reliable and robust anomaly detection systems resilient to adversarial conditions. By learning representations inherently robust to input perturbations and distribution shifts, COBRA enables safer anomaly detection deployment in security-sensitive areas where an adversary may attempt evasion. Notably, COBRA achieves high robustness without requiring anomaly data during training, valuable when such data is limited due to privacy/safety concerns. Overall, we believe COBRA importantly enhances the safety and reliability of anomaly detection systems. However, this powerful technology must be responsibly developed and deployed with technical safeguards, policy measures and institutional controls to maximize societal benefit while mitigating potential misuse risks. We advocate future work exploring robust machine learning trustworthy real-world deployment.

## P  LIMITATIONS

**Scope of Application** In this study, we primarily focus on anomaly detection for texture-based defects, which are common in real-world applications such as industrial defect detection and medical image diagnosis. Specifically, while our experiments include one-class classification (semantic anomaly detection), our performance is more pronounced in texture-based anomaly detection. In semantic anomaly detection, normal samples and anomalous samples are semantically different.

**Clean Performance** This study aims to improve the adversarial detection performance of anomaly detection tasks. Despite significant improvements in adversarial detection, our clean performance lags behind existing state-of-the-art detection methods. The trade-off between clean and adversarial test performance is well-documented in the literature Tsipras et al. (2018); Zhang et al. (2019); Madry et al. (2017); Schmidt et al. (2018); Raghunathan et al. (2020). Our work is also subject to such trade-offs. However, we have also provided results for scenarios where adversarial training is not performed.

