# OpenReview forum: "Adversarially Robust Anomaly Detection through Spurious Negative Pair Mitigation"
_ICLR.cc/2025/Conference — ICLR 2025 Poster_

### Official Review · Reviewer_mZoT · 2024-10-27

**Soundness:** 2
**Presentation:** 2
**Contribution:** 2
**Rating:** 6
**Confidence:** 3

**Summary:**

This paper introduces a novel approach for robust Anomaly Detection (AD) under adversarial attacks by generating pseudo-anomaly groups from normal group samples. It demonstrates that adversarial training with contrastive loss could serve as an ideal objective function, as it creates both inter- and intra-group perturbations. It defines opposite pairs and adversarially pulls them apart to strengthen inter-group perturbations to overcome the effect of spurious negative pairs. Experimental results demonstrate superior performance of the proposed method in both clean and adversarial scenarios, with a 26.1% improvement in robust detection across various challenging benchmark datasets.

**Strengths:**

1. It proposes COBRA (anomaly-aware COntrastive-Based approach for Robust AD), a novel method that mitigates the effect of spurious negative pairs to learn effective perturbations.

2. It employs numerous strong adversarial attacks for robustness evaluation, including PGD-1000, AutoAttack, and Adaptive AutoAttack.

3. The experiments span various datasets, including large and real-world datasets such as Autonomous Driving, ImageNet, MVTecAD, and ISIC, demonstrating COBRA’s practical applicability. Additionally, It conducts ablation studies to examine the impact of various COBRA components.

**Weaknesses:**

1. It might be too expensive to compute the anomaly score since it needs to compute the similarity score over the entire training dataset $D_{train}$. Also, it is unclear how $D_{train}$ will affect the performance. It can provide an empirical analysis of how the size of $D_{train}$ affects both performance and inference time.

2. Although it evaluates several strong attacks including AutoAttack, Adaptive AutoAttack and black-box attacks, it doesn’t evaluate adaptive attacks that are specially designed for the proposed anomaly score. It should follow the guidelines in [1] to design such adaptive attacks.

3. From Table 4, it seems $A^3$ is weaker than PGD-1000, which is not expected. Some explanations are needed here. For example, it can provide a detailed analysis of why this result occurs, including potential hypotheses about the interaction between $A^3$ and the proposed method.

[1] Tramer, Florian, et al. "On adaptive attacks to adversarial example defenses." Advances in neural information processing systems 33 (2020): 1633-1645.

**Questions:**

1. Could you analyze the computational complexity of the proposed anomaly score? Will different $D_{train}$ affect the performance?

2. Could you evaluate adaptive attacks that are specially designed for the proposed anomaly score?

3. Could you explain why $A^3$ is weaker than PGD-1000 in Table 4?

---

> ### Author Response · Authors · 2024-11-24
>
> Dear Reviewer mZoT,
>
> Thank you for your valuable comments. Here are our responses:
>
>
>
>
> >**Q1 & W1:**
>
> **Analysis of Training Data Size on Performance:**
>
> Our experiments encompass various configurations, and the appendix provides comprehensive details about the datasets and settings used (please see **Appendix L** for complete dataset details). The information relevant to this concern is already available in the manuscript, but to address the reviewer’s specific concern, we have provided additional empirical details here.
>
> As the table below indicates, there is no clear correlation between the training dataset size and the performance of our method. This demonstrates the robustness of our approach, as it performs consistently across datasets with varying sizes and resolutions.
>
>
>
> ||ImagenNet|MVTechAD|VisA|CityScapes|DAGM|ISIC2018|CIFAR10|CIFAR100|CIFAR10/CIFAR100|CIFAR10/Imagenet|
> |-|-|-|-|-|-|-|-|-|-|-|
> |Performance |85.2 / 57.0|89.1 / 75.1|75.2 / 73.8|81.7 / 56.2|82.4 / 56.8|81.3 / 56.1|83.7 / 62.3|76.9 / 51.7|81.9 / 61.2|85.5 / 53.1|
> |Training Dataset Size|30,000|3629|8659|600|5760|6,705|5,000|5,000|50,000|50,000|
> |Images Resolution|224X224|224X224|224X224|224X224|224X224|224X224|32X32|32X32|32X32|224X224|
>
>
>
> **Inference Time:**
>
>
> After training, we freeze our model and compute the features of the training samples to create a feature bank. This bank maps images to a low-dimensional embedding space (e.g., reducing dimensions from $ (1000, 3, 224, 224)$ to $(1000, 256) $. This process is performed once and does not need to be repeated.
>
> During inference, we compute the features of each test sample using the frozen model and compare its similarity to the precomputed feature bank. This is computationally efficient because the operations are performed by the frozen model in a low-dimensional space. Leveraging similarity in an embedding space for detection is a well-established and widely used approach in the literature, known for its low computational and resource requirements. Additionally, we provide detailed information about the computational times in our experiments, which are further elaborated in \textbf{Appendix Section I} of the paper.
>
>
>
> The average computational times for our model are as follows:
>
>
> * Embedding computation: 1--3 ms
> * Matrix multiplication: 1--2 ms
> * Finding the maximum value: $<$1 ms
>
> We also provide the evaluation times for competing methods for a fair comparison.
>
> ||CSI|	MSAD|	Transformaly|	ReContrast|	OCSDF|	COBRA(Ours)|
> |-|-|-|-|-|-|-|
> Per-Image Eval Time|4 ms|4 ms|5 ms|5 ms| 3 ms |4 ms|
>
> The table results are averaged over 100,000 inferences.
>
>
> >**Q2 & W2:**
>
> All adversarial attacks against anomaly detection models in our study are targeted and adaptive, designed specifically to degrade performance. Detailed explanations can be found in the Preliminaries section and Appendix K (please kindly see lines 108–119 and 1350–1364).
>
> Specifically, anomaly detection methods rely on an indicator referred to as the anomaly score, which is used to distinguish normal test samples from anomalous ones by assigning likelihoods. Ideally, an effective anomaly detector produces non-overlapping score distributions for normal and anomalous samples—assigning higher anomaly scores to anomalous samples and lower scores to normal ones.
>
> In our evaluation, we employed targeted adversarial attacks that directly manipulate the anomaly score. These attacks operate end-to-end and are specifically designed to force misclassifications by altering the score distribution. Depending on the test sample's label (normal or anomaly), the attacks aim to either maximize or minimize the anomaly score:
>
> * For anomaly test samples: The attack minimizes the anomaly score, causing the detector to incorrectly classify anomaly samples as normal (false negatives).
>
> * For normal test samples: The attack maximizes the anomaly score, causing the detector to incorrectly classify normal samples as anomalous (false positives).
>
> By targeting the anomaly score directly, these attacks effectively exploit the decision boundary between normal and anomalous predictions, inducing the most detrimental types of misclassifications for the detection system. This approach ensures that the attacks are both highly specific and impactful in the context of anomaly detection.

---

> ### Author Response · Authors · 2024-11-24
>
> >**Q3&W3:**
>
> The question raised concerns a typographical error in our manuscript, where we mistakenly referred to A³ (Adaptive Auto Attack) as A² (Auto Attack) in the column title. We sincerely apologize for this oversight. This error has been corrected in the revised version of the manuscript, and the updated results, reflecting this correction, are as follows:
>
>
>
> |Dataset|Clean|A3|AutoAttack|PGD-1000|
> |-|-|-|-|-|
> |CIFAR10|83.7|60.7|65.9|62.3|
> |CIFAR100|76.9|50.1|54.8|51.7|
> |FMNIST|93.1|90.8|87.4|89.6|
> |ImageNet|85.2|61.8|59.4|57.0|
> |MVTecAD|89.1|74.8|77.0|75.1|
> |VisA|75.2|71.6|74.9|73.8|
>
>
>
> Nevertheless, the question remains as to why PGD-1000 outperforms AutoAttack in our study. To address this, we kindly refer the reviewer to Appendix K of our manuscript, where we provide a detailed explanation of the adjustments made to AutoAttack for the anomaly detection task.
> Specifically, AutoAttack is an ensemble of six attack methods:
> 1. APGD with Cross-Entropy loss,
> 2. APGD with Difference of Logits Ratio (DLR) loss,
> 3. APGDT,
> 4. FAB \cite{croce2020minimally},
> 5. Multi-targeted FAB \cite{croce2020reliable}, and
> 6. Square Attack \cite{andriushchenko2019square}.
>
>
> Among these, the DLR loss-based attacks assume that the target model is a classifier trained on more than two classes. In the context of anomaly detection, the problem is closer to binary classification, as we are only distinguishing between normal and anomaly classes. Therefore, we excluded DLR-based attacks in our adaptation of AutoAttack and instead utilized an ensemble of the remaining attacks.
> We believe this modification is the main reason for the observed performance difference between AutoAttack and PGD-1000 in our study.

---

> > ### Author Response · Authors · 2024-11-26
> >
> > Dear Reviewer mZoT,
> >
> > we have aimed to thoroughly address your comments. If you have any further concerns, we would be most grateful if you could bring them to our attention, and we would be pleased to discuss them.
> >
> > Sincerely, The Authors

---

> > > ### Comment · Reviewer_mZoT · 2024-11-26
> > >
> > > Thanks for the detailed responses! My concerns have been addressed and I will raise my scores.

---

> ### Author Response · Authors · 2024-11-27
>
> Dear Reviewer mZoT,
>
>
>
> Thank you for your valuable review and positive feedback! We are pleased to hear that your concerns have been addressed.
>
> Sincerely, The Authors

---

### Official Review · Reviewer_qBmY · 2024-11-01

**Soundness:** 3
**Presentation:** 2
**Contribution:** 3
**Rating:** 6
**Confidence:** 3

**Summary:**

The goal of this paper is to enhance the robustness of anomaly detection systems against adversarial attacks. It introduces a new method called COBRA, which addresses the issue of spurious negative pairs and strengthens the distinction between normal and anomalous samples through contrastive learning.

**Strengths:**

1. The paper presents an approach to enhance the robustness of anomaly detection methods, specifically addressing spurious negative pairs—an issue that traditional contrastive learning methods have not adequately tackled.
2. Figure 1 is well-crafted, providing a clear visualization of the COBRA method.
3. The paper demonstrates extensive experimentation across a variety of datasets, covering both high-resolution and low-resolution data.

**Weaknesses:**

1. The claim that spurious negative pairs weaken inter-group perturbation in adversarial contrastive learning (Introduction, lines 66-69) requires supporting experimental evidence. The paper lacks clear results or a detailed explanation to substantiate this point.

2. The experiments rely solely on ResNet-18. Evaluating the method with different architectures or feature extractors would provide better insight into the generalizability of the approach.

3. The rationale behind using a k-class classifier for anomaly crafting is unclear. The paper does not sufficiently explain how training with k-class classifiers improves anomaly crafting or what the specific benefits are.

4. There is a lack of experimentation with different epsilon values during adversarial training (in the paper, a fixed epsilon of 2/255 is used for high-resolution data, and 4/255 for low-resolution data). Testing across a range of epsilon values would offer a more comprehensive evaluation of robustness.

5. The balance between clean accuracy and robust accuracy is not well-addressed. For example, in Table 2, the ZARND method shows both high clean accuracy and robust accuracy in the Low Res experiment, but the balance between these is not analyzed in depth.

6. The criteria for hard/mild augmentations are unclear. The paper does not provide a clear definition or justification for the augmentation types used.

7. The distinction between adversarial effects and augmentation effects is insufficiently explained. It’s unclear whether performance improvements are due to adversarial training or augmentations alone. This could be clarified through additional analysis.

8. Table 6 lacks sufficient explanation. Terms such as "L opposite" are not clearly defined, and the application of these terms within the experiments is not fully explained.

**Questions:**

1.  Could you please provide experimental evidence to support the claim that spurious negative pairs weaken inter-group perturbations in adversarial contrastive learning? (Introduction, lines 66-69)

2. Could you explain why ResNet-18 was chosen as the sole architecture for your experiments? It would be helpful to understand whether your proposed method generalizes well to other architectures or feature extractors, and why this aspect was not explored.

3. What is the rationale behind using a k-class classifier for anomaly crafting? It would be beneficial to know how this approach specifically improves the anomaly crafting process compared to alternative methods.

4. Could you elaborate on why different epsilon values were not explored during adversarial training? Wouldn’t varying the epsilon values offer further insights into the robustness of your method?

5. What criteria did you use to define hard and mild augmentations? It would be helpful if you could provide more details on how these augmentations impact the model’s performance.

6. How do you distinguish the effects of adversarial training from those of augmentations (Hard/Mild) ? Additional experiments clarifying the contributions of each would be greatly appreciated.

7. Could you clarify the terms used in Table 6, such as "L_{Opposite}" ? It would be valuable to understand how these terms were applied in the experiments and their significance.

Additionally, I would appreciate responses to the above weaknesses points.

---

> ### Author Response · Authors · 2024-11-23
>
> Dear Reviewer qBmY,
>
> Thanks for your constructive and valuable comments on our paper. Below, we provide a detailed response to your questions and comments. If any of our responses fail to address your concerns sufficiently, please inform us, and we will promptly follow up.
>
>
> >**W1&Q1:**
>
>
> In Table 6 of our paper , we present an ablation study comparing different loss functions, including the standard Contrastive Loss (CL) and our proposed loss. In this experiment, all components remain the same except for the objective function. The primary distinction between the standard CL and our proposed objective lies in mitigating spurious negative pairs by assigning higher weights to crafted opposite pairs.
>
> The results clearly demonstrate that the inclusion of spurious negative pairs in standard CL weakens the model's robustness against adversarial attacks. By emphasizing opposite pairs and as a result reducing the impact of spurious negatives, COBRA enhances inter-group perturbations, leading to improved robustness.
>
> Moreover, in Figure 3 of our paper, we provide t-SNE visualizations of the learned feature embeddings for both the standard CL and COBRA-trained models:
>
> Standard CL Model: The embeddings of normal and anomaly samples are not well-separated, showing significant overlap between the two groups. This indicates that the model struggles to effectively distinguish normal samples from anomalies under adversarial perturbations.
>
> COBRA Model: The embeddings exhibit a clear separation between normal and anomaly samples. The margin between the two groups is significantly larger, demonstrating enhanced inter-group discrimination and better robustness.
>
> These visualizations qualitatively support our claim that spurious negative pairs in standard CL hinder the model's ability to enforce a strong margin between normal and anomaly distributions.
>
> These visualizations qualitatively support our claim that spurious negative pairs in standard CL hinder the model's ability to enforce a strong margin between normal and anomaly distributions.
>
> Additionally, previous studies, such as  [1], have also addressed similar challenges in CL by redefining the pairing mechanism. These works underscore the limitations of standard CL in specific scenarios.
>
>
>
> *Note that, our study is distinct from [1] in its focus and application. The key difference between COBRA and them lies in the nature of the tasks they address. COBRA is designed for an open-set task, which involves detecting anomalies relative to a normal group. In contrast, FNC is tailored to a closed-set task, clustering the normal group (training set) into various semantic subgroups.*
>
>
>
>
> [1] https://openaccess.thecvf.com/content/WACV2022/papers/Huynh_Boosting_Contrastive_Self-Supervised_Learning_With_False_Negative_Cancellation_WACV_2022_paper.pdf
>
> >**W2&Q2:**
>
> We appreciate the reviewer’s valuable suggestion and understand the concern regarding the use of a single architecture in our experiments. To address this, we conducted additional experiments by replacing ResNet-18 with other widely used architectures while keeping all other components of the methodology fixed. Below, we present the results of these experiments across various datasets.
>
>
>
> *For each dataset, we report results in the “Clean / PGD-1000” format, reflecting performance under both clean and adversarial settings.*
>
> ||ImagenNet|MVTechAD|VisA|CityScapes|DAGM|ISIC2018|CIFAR10|CIFAR100|Average|
> |-|-|-|-|-|-|-|-|-|-|
> |Resnet50        |86.1 / 53.9|89.7 / 75.2|74.7 / 73.0|82.4 / 54.9|81.2 / 56.5|82.4 / 57.1|84.0 / 63.2|76.5 / 52.1|82.1 / 60.7|
> |MobileNetV2     |84.5 / 54.2|87.9 / 76.1|76.2 / 72.1|78.7 / 55.9|83.0 / 55.9|80.6 / 53.7|85.1 / 62.9|73.4 / 52.0|81.2 / 60.4|
> |PreActResNet50  |83.1 / 55.8|91.2 / 73.9|75.8 / 74.2|82.0 / 57.2|79.9 / 55.4|83.5 / 54.2|53.8 / 63.7|75.6 / 54.3|78.1 / 61.1|
> |Efficientnet_b5 |85.4 / 56.6|88.7 / 74.0|75.8 / 71.3|80.5 / 57.6|84.4 / 54.3|81.8 / 55.8|82.9 / 61.5|74.7 / 53.3|81.8 / 60.5|
> |Wide-ResNet101-2|86.1 / 58.1|90.2 / 76.0|76.1 / 72.9|82.4 / 56.9|81.1 / 57.7|80.9 / 54.9|83.1 / 59.0|76.1 / 52.7|82.0 / 61.0|
> |ResNet-18 (Ours)|85.2 / 57.0|89.1 / 75.1|75.2 / 73.8|81.7 / 56.2|82.4 / 56.8|81.3 / 56.1|83.7 / 62.3|76.9 / 51.7|81.9 / 61.2|
>
>
> While the results show slight variations across architectures, the overall trends confirm that COBRA generalizes well to other feature extractors.

---

> ### Author Response · Authors · 2024-11-23
>
> >**W3&Q3:**
>
>
> To apply a filter to the crafted pseudo-anomalies, a knowledgeable model is required. Notably, in our problem setting, no pre-trained models or additional data are available. The training set is limited to a single semantic class (e.g., car images) without any supplementary information. This constraint motivates the need to extract meaningful representations from normal data using a self-supervised approach.
>
> For instance, one potential strategy is to train an autoencoder on the normal data with a reconstruction loss and use its embeddings. However, representative studies [1,2] suggest that training a k-class classifier to predict transformations is a simple, more effective, and promising approach for representation learning in one-class classification compared to alternative methods.
>
>
> We adopt this approach, leveraging the learned embeddings from the classifier to compute the likelihood of input test samples and define a threshold for false anomaly data filtering.
>
> The **primary advantage** of our approach is that, even without utilizing additional data or external models, we propose a method to define a threshold for crafting data that confidently does not belong to the inlier set, thereby enabling the generation of effective pseudo-anomalies. It is worth noting that many existing methods [3-9] aim to craft anomalies for various tasks but often rely on extremely large generators (e.g., Stable Diffusion) or massive datasets (e.g., LAION-5B). Despite their complexity, these methods underperform compared to our proposed simple and efficient strategy for anomaly crafting. For further details, please refer to Figure 2 and Table 5.
>
>
>
>
> [1] Deep Anomaly Detection Using Geometric Transformations Izhak Golan, Ran El-Yaniv Neurips
>
> [2] Using Self-Supervised Learning Can Improve Model Robustness and Uncertainty Dan Hendrycks, Mantas Mazeika, Saurav Kadavath, Dawn Song ICLR
>
>
> [3] Lee et al, Training Confidence-calibrated Classifiers for Detecting Out-of-Distribution Samples, ICLR 2018.
>
> [4] Kirchheim et al, On outlier exposure with generative models, NeurIPS ML Safety Workshop, 2022
>
> [5] Du et al, Learning what you don’t know by virtual outlier synthesis. ICLR
>
> [6] Tao et al. Non-parametric outlier synthesis. ICLR 2023
>
> [7] Du et al, Dream the Impossible: Outlier Imagination with Diffusion Models, Neurips 2023
>
> [8]Chen et al, ATOM: Robustifying Out-of-distribution Detection Using Outlier Mining
>
> [9] RODEO: Robust Outlier Detection via Exposing Adaptive Out-of-Distribution Samples ICML 2024

---

> ### Author Response · Authors · 2024-11-23
>
> >**W4&Q4:**
>
>
> We found that it is common in the literature to fix the training $\epsilon$ and evaluate robustness across a range of $\epsilon$ values during testing (refer to Table 14), as we have already done in the manuscript (please see Table 14).
>
>
>
> To further address the reviewer's concern, we conducted additional experiments where the training $\epsilon$ was varied to assess its impact on robustness while keeping the test-time $\epsilon$ fixed.
>
>
> In this experiment, the model was trained using the default epsilon ($\epsilon$) values as specified in the manuscript: $\epsilon = \frac{4}{255}$ for low-resolution images and $\epsilon = \frac{2}{255}$ for high-resolution images. During inference, however, we varied the epsilon for adversarial testing from $0$ (clean images) up to $\frac{8}{255}$. The following table summarizes the model's performance on various datasets under different levels of adversarial perturbations:
>
> |Epsilon|ImageNet|MVTec|VisA|CityScapes|ISIC2018|CIFAR10|FMNIST|Average|
> |-------|--------|-----|----|----------|--------|-------|------|-------|
> |0|85.2|89.1|75.2|81.7|81.3|83.7|93.1|84.2|
> |$\frac{1}{255}$|68.1|81.6|74.6|65.7|62.8|77.0|91.1|74.4|
> |$\frac{2}{255}$|57.0|75.1|73.8|56.2|56.1|72.9|90.2|68.7|
> |$\frac{3}{255}$|54.1|64.7|67.2|50.9|51.8|69.3|89.9|64.0|
> |$\frac{4}{255}$|45.2|56.1|52.9|47.1|49.3|62.3|89.6|57.5|
> |$\frac{5}{255}$|43.7|49.5|46.3|40.9|45.5|60.8|89.2|53.7|
> |$\frac{6}{255}$|39.4|42.8|40.7|38.6|42.1|56.3|89.2|49.9|
> |$\frac{7}{255}$|34.5|38.1|33.9|35.2|39.6|51.8|89.1|46.0|
> |$\frac{8}{255}$|28.6|34.2|30.7|32.8|35.5|47.4|89.1|42.6|
>
>
> To address the reviewer's concern, we conducted new experiments where we varied the epsilon ($\epsilon$) value used for adversarial training, while keeping the evaluation configuration fixed (i.e., $\epsilon = \frac{4}{255}$ for low-resolution images and $\epsilon = \frac{2}{255}$ for high-resolution images). We experimented with different $\epsilon$ values ranging from $0$ (standard training without adversarial examples) to $\frac{8}{255}$. For each $\epsilon$ value used during training, we evaluated the model on both clean images and adversarial images generated with the fixed evaluation $\epsilon$. The results are summarized in the following table, where each cell shows "clean / PGD-1000":
>
>
> |Epsilon|ImageNet|MVTec|VisA|CityScapes|ISIC2018|CIFAR10|FMNIST|Average|
> |-------|--------|-----|----|----------|--------|-------|------|-------|
> |$\frac{0}{255}$|92.7 / 4.9 |95.2 / 5.8 |78.1 / 2.3 |91.7 / 3.0 |83.4 / 4.2 |94.3 / 1.7 |93.4 / 3.8 |89.8 / 3.7|
> |$\frac{1}{255}$|87.6 / 46.6|90.4 / 62.7|76.8 / 63.5|84.9 / 47.2|82.7 / 47.6|88.7 / 42.6|86.6 / 40.7|85.4 / 50.1|
> |$\frac{2}{255}$|85.2 / 57.0|89.1 / 75.1|75.2 / 73.8|81.7 / 56.2|81.3 / 56.1|86.1 / 48.7|82.7 / 44.1|83.0 / 58.7|
> |$\frac{3}{255}$|82.9 / 58.2|87.6 / 76.5|73.9 / 73.6|79.5 / 56.9|80.5 / 55.9|85.3 / 54.8|79.0 / 47.8|81.2 / 60.5|
> |$\frac{4}{255}$|81.4 / 58.9|85.4 / 75.6|73.5 / 72.2|77.0 / 57.1|79.4 / 57.0|83.7 / 62.3|76.9 / 51.7|79.6 / 62.2|
> |$\frac{5}{255}$|82.3 / 59.3|83.6 / 76.8|72.8 / 73.7|75.6 / 57.4|77.5 / 58.1|81.6 / 63.4|75.6 / 52.0|78.4 / 63.0|
> |$\frac{6}{255}$|80.5 / 58.4|83.7 / 78.2|71.4 / 74.8|75.0 / 58.0|76.7 / 57.8|82.0 / 62.9|74.2 / 53.0|77.6 / 63.3|
> |$\frac{7}{255}$|78.7 / 61.4|81.9 / 77.6|72.0 / 74.0|73.8 / 59.1|74.0 / 59.2|79.1 / 64.7|73.9 / 52.6|76.2 / 64.1|
> |$\frac{8}{255}$|77.3 / 60.2|80.9 / 78.5|70.1 / 74.1|74.9 / 58.9|73.9 / 59.9|78.9 / 65.4|73.5 / 53.7|75.6 / 64.4|
>
>  >**W5:**
>
>
>
> We acknowledge the reviewer's concern regarding the trade-off between clean and robust performance, and we will address this more thoroughly in our manuscript.
>
> However, we would like to emphasize certain points about comparing ZARND and our method, which we have also discussed in the manuscript (please see Line 890).
>
> ZARND utilizes an adversarially pretrained model on a large dataset (i.e., ImageNet), whereas our approach achieves superior performance without relying on any additional datasets or pretrained models.
>
> While we agree that on smaller datasets, such as CIFAR-10, the differences in performance between our method and ZARND may appear minor, our primary focus in this study is on real-world and industrial datasets, such as MVTecAD. On such datasets, our method demonstrates over a 45% improvement compared to ZARND. It is important to note that ZARND's performance on natural datasets benefits from its pretrained weights on ImageNet but falls short on more challenging benchmark datasets, such as MVTecAD, as discussed in our manuscript.

---

> ### Author Response · Authors · 2024-11-23
>
> >**Q5&W6:**
>
> We first review our hard augmentation (aug) strategy: We define a set of hard transforms, $T = T_i$, with each $T_i$ representing a specific type of hard aug. For each hard transformation process, which aims to shift an inlier sample to an outlier, a random subset of $k$ members from $T$ is selected, $T_{j_1}, T_{j_2}, \dots, T_{j_k}$, and sequentially applied, resulting in $T_{j_k}(\dots(T_{j_2}(T_{j_1}(x))))$. We avoid using $k = 1$, which would apply only a single hard transformation because in some cases, it does not significantly alter the semantics. For instance, applying rotation to a *Car* image yields an OOD sample as it is rare in natural images. However, some semantics are rotation invariant, such as `*Flower* images. Therefore, we have used $k > 1$. This ensures that the output of this process is sufficiently shifted from the in-distribution.
>
> For creating the set of hard augs., we include Jigsaw, Random Erasing, CutPaste, Rotation, Extreme Blurring, Intense Random Cropping, Noise Injection, Extreme Cropping, Mixup, Cutout, CutMix, and Elastic Transform. These transformations have been extensively investigated in various areas of the literature (e.g., self-supervised learning) and have been shown to be harmful for preserving semantics, often resulting in a significant shift from the original transformation. Two criteria were used for selection of such augs as hard transformations:
>
> * Several studies in OOD detection explore the effect of using augs for crafting outlier samples to provide info about OOD for detector
> models. After reviewing such works, which have provided independent studies on each hard transformation, we provided a unified view of such transformations and created a set of them instead of just relying on one [1,2,3,4,5,6].
>
> * Moreover, there have been several studies in self-supervised learning research exploring suitable augs for crafting positive pairs from each image and using them in contrastive loss (e.g., InfoNCE loss). Specifically, they choose soft augs that preserve semantics (e.g., mild color jitter). Another rationale behind creating such a set of hard transformations is that we choose transformations that the literature shows are harmful for preserving semantics and avoided using them (e.g., rotation) [7,8,9].
>
> * For mild/soft augmentations in training step, we follow common practices in the self-supervised learning literature (e.g., color jitter) and refrain from making any modifications.
>
> [1] Tack et al. CSI 2020
>
> [2] Kalantidis et al.Hard negative 2020
>
> [3] Li et al.Cutpaste 2021
>
> [4] Sinha et al.Negative data 2021
>
> [5] Miyai et al.Rethinking 2023
>
> [6] Zhang et al.Improving the 2024
>
> [7] Chen et al.SimCLR 2020
>
> [8] Grill et al.BYOL
>
> [9] He et al.MOCO
>
> ---
>
> >**Q7&W8:**
>
> We sincerely apologize for the issue, which appears to have resulted from a typographical error in our manuscript. Some equations in the definition were inadvertently omitted, and we have now revised the manuscript to address this oversight (please see line 285). Specifically, after proposing our loss function, we intended to provide an intuition for the objective behind our approach.
>
> We initially claimed that in the case of adv. training with $L_{CL}$, the objective is to ensure that the embeddings of both the normal and pseudo-normal groups become compact. This is achieved by weighting positive pairs to bring them closer together.
>
> Conversely, adv. training with $L_{\text{Opposit}}$ was defined to offer better insight into the mechanics of $L_{\text{COBRA}}$ rather than to serve as a standalone recommended loss function. Specifically, $L_{\text{Opposit}}$ explicitly pushes the distributions of the normal and anomaly groups further apart.
>
> Intuitively, adv. training with  $L_{\text{COBRA}}$ integrates these two objectives, producing embeddings that are compact within each group while maximizing the separation between the embeddings of the two groups. The ablation study presented in Table 6 demonstrates that neither $L_{CL}$ nor $L_{\text{Opposit}}$ alone achieves the performance of $L_{\text{COBRA}}$. This underscores the importance of combining both properties: compact representations for the normal and anomaly groups and maximizing their separation.
>
> $L_{\text{COBRA}}= \sum \sum  \log \frac{\exp(\text{sim}(f(x_i), f(x_j))/t) - \exp(\text{sim}(f(x_i), f(\Upsilon(x)))/t)}{\sum\limits_{x_k \in \text{P}(x_i) \cup \text{N}(x)} \exp(\text{sim}(f(x_i), f(x_k))/t)},$
>
>
>
>
>
> $L_{\text{Opposite}}= \sum \sum  \log \frac{ - \exp(\text{sim}(f(x_i), f(\Upsilon(x)))/t)}{\sum\limits_{x_k \in \text{P}(x_i) \cup \text{N}(x)} \exp(\text{sim}(f(x_i), f(x_k))/t)},$
>
>
> $ L_{\text{CL}}= \sum \sum  \log \frac{\exp(\text{sim}(f(x_i), f(x_j))/t) }{\sum\limits_{x_k \in \text{P}(x_i) \cup \text{N}(x)} \exp(\text{sim}(f(x_i), f(x_k))/t)},$
>
> $L_{\text{COBRA}} \simeq L_{\text{CL}}+ L_{\text{Opposite}}.$

---

> ### Author Response · Authors · 2024-11-23
>
> >**W7 & Q6-(1):**
>
> In response to the reviewer’s question regarding the role of transformations and adversarial training, we will first review our method and then directly address the concerns raised. To clarify, the use of hard transformations (along with threshold-based filtering) in our pipeline is specifically designed to create pseudo-anomalies, which serves a purpose distinct from that of mild augmentations in our approach. Mild augmentations, commonly used in self-supervised learning, are a standard practice in CL domain,  that we also leverage during training with $L_{COBRA}$.
>
> ---
>
> ## Motivation for designing robust anomaly detection
>
>
> Developing robust anomaly detection methods is crucial due to their application in safety-critical real-world problems such as autonomous driving. While clean results on challenging benchmarks (e.g. MVTecAD) achieve near 100% AUROC, the best performance of previous works under attacks is less than random detection even on tiny datasets. Motivated by this, we propose COBRA, an adversarially robust model that improves robust detection by 26.1%. COBRA is based on three components: using **pseudo-anomaly samples**, **adversarial training**, and introducing a **novel loss function**, which we explain in more detail below.
>
>
>  ## Motivation behind COBRA
> We hypothesized that for effective robust anomaly detection, the model must be exposed to both anomalies and normal samples during training. Otherwise, the detector can be easily fooled by perturbations on anomaly test samples during inference. This hypothesis aligns with observations from related works [1,2,3].
>
> However, in anomaly detection setups, abnormals are unavailable [16,17,18]. Moreover, collecting extra abnormal datasets presents challenges, such as the need to process and remove normals to avoid providing misleading information for the detector. Additionally, extra data may bias the detector model, and obtaining relevant datasets, especially for medical imaging, is costly. As a result, we propose a novel approach for crafting pseudo-anomaly samples. We use hard transformations such as rotation and cut-paste on normals, which have shown to shift normals to an anomaly distribution [9,10,14]. To ensure transformed samples do not still belong to the normal distribution based on the characteristics of the dataset, we develop a threshold mechanism. Adversarial training and its variants have been shown to be the most effective defense for DNNs. Thus, we chose adversarial training as a core component of our method, specifically utilizing the common adversarial training approach, PGD-10.
>
> We also explored the impact of different loss functions on the robustness of anomaly detection. A robust detector should achieve separated embeddings for normal and anomaly groups. Contrastive Learning (CL) has been shown to be more effective than classification (CLS) loss for anomaly detection [14]. CL operates by attracting positive pairs and repelling negative pairs. However, using CL with adversarial training fails to achieve robust detection because negative pairs include normal-normal and anomaly-anomaly pairs, referred to as spurious negative pairs. These weaken inter-group perturbations. To address this, we propose a new loss function that explicitly increases the distance between normal and anomaly distributions by repelling opposite pairs which are confirmed to be inter-group pairs. It is important to highlight that the limitations of CL in anomaly detection are apparent in adversarial scenarios. This stems from the fact that adversarial training requires a high degree of data complexity compared to standard settings, necessitating a broad range of strong perturbations to achieve robust anomaly detection [19, 20].
>
> [1] Chen, Robust OOD, 2020
>
> [2] Azizmalayeri, Your, 2020
>
> [3] Chen, Atom, 2021
>
> [4] Kalantidis et al., Hard, 2020
>
> [5] Li Cutpaste, 2021
>
> [6] Sinha Negative Data, 2021
>
> [7] Miyai Rethinking Rotation, 2023
>
> [8] Zhang Improving, 2024
>
> [9] Chen Novelty, 2021
>
> [10] DeVries Improved, 2017
>
> [11] Yun Cutmix, 2019
>
> [12] Akbiyik Data, 2019
>
> [13] Ghiasi Copy-Paste 2020
>
> [14] Tack CSI, 2020
>
> [15] Cohen Transformaly, 2022
>
> [16] Yang Generalized, 2022
>
> [17] Salehi A Unified, 2022
>
> [18] Perera One-Class, 2021
>
> [19] Schmidt Adversarially 2018
>
> [20] Stutz Disentangling 2019
>
> [21] Golan GT 2018
>
> [22]Hendrycks Using 2019

---

> ### Author Response · Authors · 2024-11-23
>
> >**W7 & Q6-(2):**
>
>
> Based on the provided description, we interpret the reviewer's question as an inquiry into the impact of adversarial training and pseudo-anomalies in our proposed method. This interpretation is grounded in the fact that the hard transformations in our pipeline are specifically designed to craft pseudo-anomaly samples.
>
> To address this, we have conducted ablation studies to analyze the individual contributions of these components. These experiments demonstrate the effect of each component on the robustness of the proposed approach. Additionally, we note that these results are also presented in the manuscript, but we provide further discussion here to address the reviewer's concerns in more detail. It is important to emphasize that adversarial training is a critical component of our pipeline, and its combination with pseudo-anomalies and the proposed loss function achieves strong, robust performance. Removing adversarial training significantly compromises the model's ability to safeguard against anomalies.
>
> The table below summarizes the experimental setups and results across multiple datasets:
>
> *(Clean/PGD-1000)*
> || Adversarial Training | Pseudo-Anomaly |ImagenNet|MVTechAD|VisA|CityScapes|DAGM|ISIC2018|CIFAR10|CIFAR100|
> |-|--|-|-|-|-|-|-|-|-|-|
> |Setup A|❌|❌|73.1 / 2.5|60.4 / 1.6|61.7 / 0.5|71.8 / 2.6|68.7 / 3.2|67.8 / 4.1|76.8 / 0.2|69.4 / 2.5|
> |Setup B|✔️|❌|72.8 / 32.5|57.6 / 10.8|59.1 / 13.6|68.8 / 18.4|67.6 / 8.5|65.2 / 21.7|78.6 / 50.3|68.5 / 35.4|
> |Setup C|❌|✔️|92.7 / 4.9|95.2 / 5.8|78.1 / 2.3|91.7 / 3.0|90.8 / 4.8|83.4 / 4.2|94.3 / 1.7|93.7 / 4.6|
> |Setup D (default)|✔️|✔️ (**Ours**)|85.2 / 57.0|89.1 / 75.1|75.2 / 73.8|81.7 / 56.2|82.4 / 56.8|81.3 / 56.1|83.7 / 62.3|76.9 / 51.7|
>
>
>
> We also highlight that our proposed loss function, $L_{\text{COBRA}}$, requires pseudo-anomalies for training. In the absence of pseudo-anomalies, as in Setups A and B, we substitute $L_{\text{COBRA}}$ with a contrastive loss ($L_{\text{CL}}$) to enable training. This substitution is necessary to facilitate the provided experiments and analyze the role of pseudo-anomalies.

---

> ### Author Response · Authors · 2024-11-26
>
> Dear Reviewer qBmY,
>
> thank you again for your review. We wanted to check in to see if there are any further clarifications we can provide. We hope that our updated PDF, including new experiments and explanations, effectively addresses your concerns.
>
>
> Best,
> the authors

---

> ### Author Response · Authors · 2024-12-02
>
> Dear qBmY,
>
> We apologize for reaching out again.
>
>
> While we are pleased to know that your concerns have been addressed, the assigned score suggests there may still be unresolved issues. We would like to engage further to address these concerns.
>
> we hope the new experiments and revisions to the manuscript allow you to reevaluate your score.
>
> Thank you!

---

### Official Review · Reviewer_1DsR · 2024-11-02

**Soundness:** 3
**Presentation:** 3
**Contribution:** 3
**Rating:** 6
**Confidence:** 3

**Summary:**

This paper proposes a novel method for adversarially robust anomaly detection. The proposed method enhances robustness in anomaly detection by generating pseudo-anomalies through hard transformations and mitigating the effect of spurious negative samples with a tailored contrastive loss. Experiments on various datasets demonstrate that the proposed method achieves superior performance in both clean and adversarial scenarios.

**Strengths:**

1. This paper identifies limitations in existing anomaly detection methods and challenges in using  contrastive learning for adversarial training, offering solutions to improve robustness in anomaly detection.
2. Through extensive experiments, this paper provides a comprehensive understanding of the strengths and limitations of the proposed method.

**Weaknesses:**

1. This paper does not follow the ICLR citation format and contains several typos affecting clarity.
  - Page 4, line 207: In the phrase "where $\Upsilon(x^{b+i})$ is considered as $x^{b}$", it seems that $x^{b}$ should be replaced with $x^{i}$.
  - Page 6, line 291: $x+\delta$ should be used as an input to $\mathcal{L}(\cdot, \cdot)$ instead of $x+\delta^{\ast}$.
  - Page 6, lines 317 and 319: There are typos involving "setting 4".
2. The paper includes unclear or inconsistent content that hinders understanding of the proposed method.
  - Page 2, line 107: The formulation $x^{\ast}=x_{k}^{\ast}$ includes the undefined symbol $k$.
  - In Figure 1, an augmented sample and its corresponding opposite sample (e.g., $\text{A}\_1$ and $\text{A}\_1^{\Upsilon}$) are used in $\mathcal{L}\_{\text{COBRA}}$. However, in equation (2), the opposite sample of the original sample (e.g., $\text{A}^{\Upsilon}$) is used instead.
  - In Figure 1 and the “Adversarial Training Step” paragraph on page 6, a single adversarial example $x_{adv}$ is generated for the augmented samples $x_i$ and $x_j$. However, the generation process is unclear and needs a step-by-step explanation, especially on how the perturbation is optimized in $\mathcal{L}\_{\text{COBRA}}$.
3. In the EXPERIMENTS section, the analysis of experimental results appears before the experimental setup, evaluation details, and implementation details, which limits understanding. Additionally, the analysis is too brief to fully convey the implications of the experimental results, and no analysis is provided for Figure 3.

**Questions:**

1. Could the authors address the first and second weaknesses to improve the clarity of the paper?
  - Update the citation format to follow the ICLR style and fix the typos.
  - Define $k$ and resolve the inconsistency regarding the opposite sample in Figure 1 and equation (2).
  - If possible, include pseudocode for generating the adversarial example.
2. In the EXPERIMENTS section, could the authors first present the experimental setup and evaluation details, and implementation details, followed by an analysis of the results? Additionally, could the authors provide a more detailed analysis for each experimental result and include an analysis of Figure 3?
3. On page 7, lines 375-377, it states that replacing adversarial training with clean training improved clean detection performance to 90.7%. Could the authors also report how robust accuracy was affected?
4. For the PGD attack, the number of iterations was set to 1000. Does this actually increase the effectiveness of the adversarial attack? Could the authors compare the robust accuracy with fewer iterations, such as 20 or 100?

---

> ### Author Response · Authors · 2024-11-24
>
> Dear Reviewer 1DsR,
>
>
> We appreciate your constructive feedback on our manuscript. Below, we provide a detailed response to your questions and comments.
>
>
> >**W1:**
>
> Thank you for your feedback. We have carefully addressed all your suggestions and would greatly appreciate it if you could kindly see our revised manuscript.
>
>
>
> >**W2&Q1:**
>
> >**In Figure 1, an augmented sample and its corresponding opposite sample (e.g., $A_1$ and $A_1^\Upsilon$) are used in $\mathcal{L}_{\text{COBRA}}$. However, in equation (2), the opposite sample of the original sample (e.g., $A^\Upsilon$) is used instead.**
>
> Thank you for noting this point. The $A^\Upsilon$ in equation (2) could also be replaced by $A_1^\Upsilon$, as both $A$ and $A_1$ are positive pairs and share similar semantics. Applying hard transformations to these samples results in closely related pseudo-anomalous samples, a behavior we have also observed experimentally. To further clarify this point, we explicitly mention it in line 286.
>
> >**In Figure 1 and the "Adversarial Training Step" paragraph on page 6, a single adversarial example $x_{\text{adv}}$ is generated for the augmented samples $x_i$ and $x_j$. However, the generation process is unclear and needs a step-by-step explanation, especially on how the perturbation is optimized in $\mathcal{L}_{\text{COBRA}}$.&If possible, include pseudocode for generating the adversarial example.**
>
> Thank you for pointing this out. To clarify, the perturbation $x_{\text{adv}}$ is computed with respect to the entire batch $X$, rather than specifically the augmented samples $x_i$ or $x_j$. This is achieved using the PGD-10 algorithm and the defined objective function. We will work on improving Figure 1 and adding further clarification to better illustrate this point. Additionally, we would like to highlight that we already provide an extensive pseudocode on page 16, which includes the computation of adversarial perturbations. Nevertheless, we will make additional efforts to enhance the clarity of this explanation.
>
>
> >**W3&Q2:**
>
> We thank the reviewer for their insightful suggestions. In response, we have revised the structure of the EXPERIMENTS section to enhance clarity and coherence. Additionally, we will provide a more detailed analysis of Figure 3 and the experimental results in our manuscript.
>
> Figure 3 presents  visualizations of features extracted by the encoder, trained on CIFAR-10 in a one-class setup with 'Automobile' as the normal class. The visualizations compare representations learned under various loss functions: Contrastive Loss (CL), Classification Loss (CLS), Supervised Contrastive Loss (SupCL), and the proposed COBRA method, while keeping other components constant.
>
> The COBRA method demonstrates superior performance, achieving compact clustering for normal samples and a significant separation from pseudo-anomalies, even under adversarial perturbations. In comparison, alternative loss functions such as CL, SupCL, and CLS fall short in these aspects.
>
>  >**Q3:**
> To address the reviewer’s concern, we present the results for both scenarios : with and without adversarial training. Notably, replacing adversarial training with standard training leads to improved clean performance, albeit at the cost of reduced robustness. This demonstrates that, in environments where reliability is guaranteed and adversarial agents are absent, our method provides the flexibility to prioritize clean performance over robustness. This adaptability is a key feature of the COBRA framework, enabling it to be tailored to the specific requirements of the deployment environment.
>
>
>
> *(Clean/PGD-1000)*
> || Adversarial Training  |ImagenNet|MVTechAD|VisA|CityScapes|DAGM|ISIC2018|CIFAR10|CIFAR100|
> |-|-|-|-|-|-|-|-|-|-|
> | |❌|92.7 / 4.9|95.2 / 5.8|78.1 / 2.3|91.7 / 3.0|90.8 / 4.8|83.4 / 4.2|94.3 / 1.7|93.7 / 4.6|
> |  (default)|✔️|85.2 / 57.0|89.1 / 75.1|75.2 / 73.8|81.7 / 56.2|82.4 / 56.8|81.3 / 56.1|83.7 / 62.3|76.9 / 51.7|
>
>
>  >**Q4:**
>
>
> For adversarial training, as noted in the manuscript, we use PGD with 10 steps. However, for adversarial testing, we employ PGD-1000 to ensure the attack is sufficiently strong. To address the reviewer’s concern, we have conducted additional experiments where all other components are fixed, and the number of PGD steps during testing is varied. The results of these experiments are provided below.
>
> As the results indicate, reducing the number of PGD steps during testing leads to a minor weakness in the adversarial evaluation, reflecting slightly less effective attacks. These findings validate our choice of using PGD-1000 for robust evaluation during testing to ensure rigor.
>
>
>
>
> ||ImagenNet|MVTechAD|VisA|CityScapes|DAGM|ISIC2018|CIFAR10|CIFAR100|MNIST|FMnist|SVHN| Average|
> |-|-|-|-|-|-|-|-|-|-|-|-|-|
> |PGD-20|59.7|77.5|74.9|58.7|60.2|61.8|63.9|54.8|95.7|91.3|61.8|69.1|
> |PGD-100|58.2|76.6|74.5|57.6|57.3|57.6|62.4|52.5|95.0|90.8|59.7|67.5|
> |PGD-1000 *(default)*|57.0|75.1|73.8|56.2|56.8|56.1|62.3|51.7|96.4|89.6|58.2|66.7|

---

> ### Author Response · Authors · 2024-11-27
>
> Dear Reviewer 1DsR,
>
> We sincerely appreciate the thoughtful feedback you’ve offered on our manuscript. We have carefully reviewed each of your comments and have made efforts to address them thoroughly. We kindly ask you to review our responses and share any additional thoughts you may have on the paper or our rebuttal. We would be more than happy to accept all your criticisms and incorporate them into the paper.
>
> Sincerely, The Authors

---

> > ### Comment · Reviewer_1DsR · 2024-11-28
> >
> > Thank you for addressing my questions and resolving the weaknesses I pointed out. I appreciate your effort and will maintain my current score.

---

> ### Author Response · Authors · 2024-12-02
>
> Thank you for your feedback. We're glad your comments have been addressed!
>
> Sincerely, The Authors

---

### Official Review · Reviewer_YXiZ · 2024-11-03

**Soundness:** 3
**Presentation:** 3
**Contribution:** 3
**Rating:** 6
**Confidence:** 4

**Summary:**

The paper addresses adversarial robustness in anomaly detection settings, where conventional models are vulnerable due to training limitations that exclude labeled data and anomaly samples. The authors propose a framework that generates pseudo-anomalies (synthetic outliers) through data augmentations and employs a contrastive loss function. They claim a modifications to the contrastive objective mitigates the effect of spurious negative pairs by maximizing the margin between normal and anomaly samples, although, it seems class collision still occurs despite their modification. The authors experiment with well-known benchmark datasets, demonstrating significant improvement in robust detection performance across real-world scenarios under adversarial conditions.

**Strengths:**

-	The paper addresses the topic of adversarial robustness in anomaly detection, which is a known limitation in semantic and defect-based anomaly detection, and unlabeled OOD detection.
-	The authors provide an extensive set of experiments, comparing their method with well-known approaches in a comprehensive manner.
-	The literature review includes relevant works.
-	The modifications proposed are straightforward and appear to yield strong results, though their simplicity raises some questions about the underlying mechanisms driving the reported improvements.

**Weaknesses:**

-	The approach to mitigating spurious negative pairs through opposite pairs is not clearly effective in addressing class collision. The use of a single-pairing strategy for forming the positives, inherited from InfoNCE/NT-Xent suggests unresolved issues.
-	The source of performance gains is unclear. It is not evident whether improvements stem from the redefinition of the loss function, thresholding, opposite pairs, the anomaly crafting strategy, etc. A clearer breakdown of these contributions would clarify the method’s impact.
-	The term "spurious pairs" seems misapplied, as it should be referred to as "class collision" in the context of contrastive learning. Using standard terminology would align the paper with accepted practices in the field. Refer to  Arora et al. (2019, "A Theoretical Analysis of Contrastive Unsupervised Representation Learning").
-	It is difficult for the reader to rapidly identify what is being considered as positive and negative samples within the formulation. A clear understanding of this within the framework is critical for the reader's comprehension.

**Questions:**

-	Does the source of performance improvements primarily come from the redefinition of the loss function (e.g., the inclusion of opposite pairs) or from the way anomalies are crafted? Ablation studies with a detailed breakdown of each part of the framework, including replacements with other contrastive objectives, could clarify this.
-	Have the authors considered techniques from recent or ongoing research that expand the set of positives for normal samples to include all other normals in the batch? This approach could potentially enhance robustness by promoting compact in-distribution representations at the same time increasing the margin between normal and anomalous samples.
-	On page 1 (line 051), the authors refer to a “novel thresholding approach (19).” It’s unclear why this is considered novel, especially since the cited work is from 2013. The authors should clarify already in the introduction what aspect of their thresholding method is novel or provide more context to justify the claim.
-	The introduction could be improved by breaking down and summarizing the key components of the methodology. Currently, it requires readers to navigate back and forth to piece together the details. The authors should use the introduction to outline their approach comprehensively, giving readers a clear understanding of what will be discussed in detail later in the methodology section.
-	On page 1 (line 57), the authors state that the "optimal objective function should maximize the margin between the distributions of normal and anomaly samples." The rationale behind how maximizing this margin, while also promoting compactness of in-distribution samples, contributes to adversarial robustness is not clearly explained. The authors should provide additional insights on why this separation is relevant and how it enhances the model's robustness against adversarial attacks.
-	In Section 3.1, the explanation regarding the generation of anomalies and the definition of opposite pairs needs to be clearer. Authors should easily introduce the definitions, for any given normal sample, what is being considered positives, which are negatives, and where/how the opposite pairs come within this settings. A more direct and clearer explanation would greatly enhance the reader's comprehension of the framework.
-	The paper refers to InfoNCE but cite SimCLR paper. Moreover, the implementation details and mathematical definitions follows more NT-Xent. Could the authors clarify whether the loss formulation adheres strictly to InfoNCE or NT-Xent? Additionally, in the context of defining negative pairs, the explanation on Page 4, Line 214, seems to align with NT-Xent, particularly since you describe the negative samples as "the other samples’ augmented view." This typically implies $2(N-1)$ negative samples per anchor, following NT-Xent. Is this interpretation correct, or does your formulation differ in the way negatives are constructed within the batch?
-	Would the authors consider reformulating Equations (1) and (2) to follow the standard mathematical notation used in InfoNCE or NT-Xent? The current two-summation format may be confusing, especially since $P(x_i)$ is a singleton. Using a more familiar notation could improve clarity and make it easier for readers to understand the loss function.

---

> ### Author Response · Authors · 2024-11-23
>
> Dear Reviewer YXiZ,
>
>
> Thank you for your constructive and valuable comments on our paper. First, we provide a brief overview of the motivation behind our method, followed by detailed responses to each of your comments.
>
>
>
> ## Motivation for designing robust anomaly detection
>
>
> Developing robust anomaly detection methods is crucial due to their application in safety-critical real-world problems such as autonomous driving. While clean results on challenging benchmarks (e.g. MVTecAD) achieve near 100% AUROC, the best performance of previous works under attacks is less than random detection even on tiny datasets. Motivated by this, we propose COBRA, an adversarially robust model that improves robust detection by 26.1%. COBRA is based on three components: using **pseudo-anomaly samples**, **adversarial training**, and introducing a **novel loss function**, which we explain in more detail below.
>
>
>  ## Motivation behind COBRA
> We hypothesized that for effective robust anomaly detection, the model must be exposed to both anomalies and normal samples during training. Otherwise, the detector can be easily fooled by perturbations on anomaly test samples during inference. This hypothesis aligns with observations from related works [1,2,3].
>
> However, in anomaly detection setups, abnormals are unavailable [16,17,18]. Moreover, collecting extra abnormal datasets presents challenges, such as the need to process and remove normals to avoid providing misleading information for the detector. Additionally, extra data may bias the detector model, and obtaining relevant datasets, especially for medical imaging, is costly. As a result, we propose a novel approach for crafting pseudo-anomaly samples. We use hard transformations such as rotation and cut-paste on normals, which have shown to shift normals to an anomaly distribution [9,10,14]. To ensure transformed samples do not still belong to the normal distribution based on the characteristics of the dataset, we develop a threshold mechanism. Adversarial training and its variants have been shown to be the most effective defense for DNNs. Thus, we chose adversarial training as a core component of our method, specifically utilizing the common adversarial training approach, PGD-10.
>
> We also explored the impact of different loss functions on the robustness of anomaly detection. A robust detector should achieve separated embeddings for normal and anomaly groups. Contrastive Learning (CL) has been shown to be more effective than classification (CLS) loss for anomaly detection [14]. CL operates by attracting positive pairs and repelling negative pairs. However, using CL with adversarial training fails to achieve robust detection because negative pairs include normal-normal and anomaly-anomaly pairs, referred to as spurious negative pairs. These weaken inter-group perturbations. To address this, we propose a new loss function that explicitly increases the distance between normal and anomaly distributions by repelling opposite pairs which are confirmed to be inter-group pairs. It is important to highlight that the limitations of CL in anomaly detection are apparent in adversarial scenarios. This stems from the fact that adversarial training requires a high degree of data complexity compared to standard settings, necessitating a broad range of strong perturbations to achieve robust anomaly detection [19, 20].
>
> [1] Chen, Robust OOD, 2020
>
> [2] Azizmalayeri, Your, 2020
>
> [3] Chen, Atom, 2021
>
> [4] Kalantidis et al., Hard, 2020
>
> [5] Li Cutpaste, 2021
>
> [6] Sinha Negative Data, 2021
>
> [7] Miyai Rethinking Rotation, 2023
>
> [8] Zhang Improving, 2024
>
> [9] Chen Novelty, 2021
>
> [10] DeVries Improved, 2017
>
> [11] Yun Cutmix, 2019
>
> [12] Akbiyik Data, 2019
>
> [13] Ghiasi Copy-Paste 2020
>
> [14] Tack CSI, 2020
>
> [15] Cohen Transformaly, 2022
>
> [16] Yang Generalized, 2022
>
> [17] Salehi A Unified, 2022
>
> [18] Perera One-Class, 2021
>
> [19] Schmidt Adversarially 2018
>
> [20] Stutz Disentangling 2019
>
> [21] Golan GT 2018
>
> [22]Hendrycks Using 2019
>
> ---
>
> >**W1:**
>
> While we adopt a single-pairing strategy for forming positives, similar to common contrastive learning (CL) frameworks, our method addresses the issue of class collision (spurious negative pairs) by introducing a new category of pairs. By emphasizing opposite pairs rather than uniformly treating all negatives, we ensure that the model focuses on maximizing the margin between normal and anomaly groups, thereby avoiding intra-class conflicts.
>
> We believe that our proposed framework effectively addresses the mentioned challenges, resulting in a robust anomaly detection method. As demonstrated in our ablation study (please kindly see Table 6), $L_{COBRA}$ outperforms $L_{CL}$ in terms of performance, with all other components remaining identical except for the objective function.

---

> ### Author Response · Authors · 2024-11-23
>
> >**Q1:**
>
> We should highlight that our study includes a comprehensive ablation study that examines the effects of each component of our method. This includes the use of extra anomalies (Table5), the loss function (Table6), and the impact of adversarial training (Table 9). To address the reviewer's concerns, we have provided those experiments here.
>
>
>
> *(Clean/PGD-1000)*
>
>
> | Loss Functions | Extra Anomaly | MVTec | VisA | ImageNet | CityScapes | ISIC2018 | CIFAR10 | FMNIST | Average |
> |-|-|-|-|-|-|-|-|-|-|
> | CL | ❌ | 57.6/10.8 | 59.1/13.6 | 67.8/32.5 | 68.8/18.4 | 65.2/21.7 | 78.6/50.3 | 82.4/71.7 | 68.5/31.3 |
> | CL | ✔️ | 76.4/58.7 | 67.8/58.5 | 68.3/47.6 | 70.1/42.8 | 72.7/44.0 | 67.9/54.3 | 88.3/82.5 | 73.1/55.5 |
> | CLS | ✔️ | 62.6/40.4 | 62.0/40.7 | 59.5/45.1 | 72.1/46.2 | 70.7/41.9 | 62.6/49.6 | 82.0/78.4 | 67.4/48.9 |
> | CL+CLS | ✔️ | 78.5/59.1 | 69.3/60.1 | 70.5/51.9 | 72.8/48.7 | 74.1/45.1 | 70.2/58.1 | 88.7/81.4 | 74.8/57.8 |
> | SupCL | ✔️ | 80.4/60.5 | 70.5/62.7 | 74.6/46.3 | 73.6/44.6 | 75.0/45.7 | 74.2/53.8 | 91.5/83.2 | 77.1/56.7 |
> |COBRA+CLS *(Ours)* | ✔️ | 89.1/75.1 | 75.2/73.8 | 85.2/57.0 | 81.7/56.2 | 81.3/56.1 | 83.7/62.3 | 93.1/89.6 | 84.2/67.2 |
>
>
>
> || Adversarial Training  |MVTechAD|VisA|ImagenNet|CityScapes|DAGM|ISIC2018|CIFAR10|CIFAR100|
> |-|-|-|-|-|-|-|-|-|-|
> | |❌|95.2 / 5.8|78.1 / 2.3|92.7 / 4.9|91.7 / 3.0|90.8 / 4.8|83.4 / 4.2|94.3 / 1.7|93.7 / 4.6|
> |  (default)|✔️|89.1 / 75.1|75.2 / 73.8|85.2 / 57.0|81.7 / 56.2|82.4 / 56.8|81.3 / 56.1|83.7 / 62.3|76.9 / 51.7|

---

> ### Author Response · Authors · 2024-11-24
>
> >**Q2:**
>
> We appreciate your thoughtful suggestion regarding expanding the set of positives for normal samples to include all other normals in the batch.
> To address the reviewer's concern, we conducted additional experiments where we fixed all other components of our method but replaced our loss function with Contrastive Loss (CL). In this setup, we treated all normal samples within a batch as positive pairs during training. The results of these experiments are provided below.
>
> However, this approach resulted in a significant decline in performance.
>
> The primary reason for this performance degradation is the occurrence of **mode collapse** and **over-clustering** within the embedding space for normal samples. Specifically:
>
>
> **Mode Collapse:** Treating all normal samples as positives forced the model to align dissimilar normal samples and map them to a similar embedding space, disregarding their inherent variability. For example, in the CIFAR-10 vs. CIFAR-100 setup (where CIFAR-10 represents the normal set), this alignment reduced the diversity of embeddings within the normal class, which in turn impaired the model’s ability to generalize to unseen normal and anomalous samples.
>
>   **Over-Clustering:** Compressing all normal samples into an overly tight cluster reduced the margin between the normal and anomaly classes. This diminished the model's ability to distinguish between normal and pseudo-anomalous samples, particularly for near-boundary anomalies.
>
> These issues negatively impacted both clean and adversarial robustness.
>
> We believe these findings highlight the critical importance of balancing compactness within the normal class with the preservation of inter-class margins.
>
>
> | Loss Functions   | MVTec | VisA | ImageNet | CityScapes | ISIC2018 | CIFAR10 | FMNIST | CIFAR-10 vs CIFAR-100|
> |-|-|-|-|-|-|-|-|-|
> | Proposed Method | 68.1/42.5 | 58.3/44.0 | 49.0/30.1 | 56.8/27.8 | 54.3/27.4 | 61.9/40.3 | 71.9/62.9 | 48.5/11.7 |
> | *(Ours)*  | 89.1/75.1 | 75.2/73.8 | 85.2/57.0 | 81.7/56.2 | 81.3/56.1 | 83.7/62.3 | 93.1/89.6 | 81.9/61.2
>
>
>
> >**Q3:**
>
> Pseudo-anomaly samples, especially those located near the normal distribution (near-anomalies), provide valuable information for anomaly detection. Many methods focus on crafting such anomalies.
> Our approach stands out because, without relying on additional data or external models, we propose a threshold-based method for crafting data that confidently lies outside the inlier set. This enables the effective generation of pseudo-anomalies.
> While existing methods [1-7] often rely on large generators (e.g., Stable Diffusion) or extensive datasets (e.g., LAION-5B), they lack thresholding or filtering strategies. This increases the risk of including false anomalies—normal samples mistakenly treated as anomalies. Despite their complexity, these methods underperform compared to our simple and efficient strategy, as shown in Figure 2 and Table 5.
>
> **Key Contributions of Our Approach:**
>
> 1. Crafting Pseudo-Anomalies from Normal Samples: Instead of utilizing external anomaly datasets, we generate anomaly samples through hard augmentations of normal data. This avoids challenges associated with incorporating real anomaly data, such as ensuring it excludes normal concepts that could mislead the detector.
>
> 2. Auxiliary Task for Effective Representation: In the absence of labels, we employ an auxiliary task for learning meaningful embeddings, using transformation prediction as the encoder.
>
> Our method demonstrates superior performance with a simpler and more efficient approach, addressing the limitations of existing anomaly-crafting techniques in domains like medical imaging, where gathering real near-anomaly samples is costly and time-consuming.
>
> 3. Threshold-Based Filtering for False Anomalies: While hard augmentations can shift normal samples toward anomaly-like regions, they may still produce falsely crafted anomalies. To mitigate this, we use a Gaussian Mixture Model (GMM) to estimate likelihoods based on the embeddings of training samples. These embeddings are extracted using the transformation predictor encoder. A likelihood-based threshold, such as the bottom 5% of the distribution, is then applied to filter out samples with a high likelihood of being normal.
>
>
> >**On page 1 (line 051), the authors refer to a “novel thresholding approach (19).” It’s unclear why this is considered novel, especially since the cited work is from 2013.**
>
> We cited the GMM paper, which presents a general and widely used classic method for modeling low-dimensional embedding spaces. It is important to note that GMM is not a filtering-based method.
>
> ---
>
> [1] Lee et al, TraininClass  , ICLR 2018.
>
> [2] Kirchheim et al, On outlier with, 2022
>
> [3] Du et al, Learning  virtual outlier synthesis. ICLR
>
> [4] Tao et al. Non-parametric outlier synthesis. ICLR 2023
>
> [5] Du et al, Dream the:   Diffusion Models, Neurips 2023
>
> [6]Chen et al, ATOM: Robustifying Mining
>
> [7] RODEO: Robust Outlier via ICML 2024

---

> ### Author Response · Authors · 2024-11-24
>
> >**Q5:**
>
> Maximizing the margin between normal and anomaly distributions creates a larger separation in the feature space, making it harder for adversarial perturbations to cause misclassification. Adversaries would require stronger perturbations to shift samples across the margin, which often exceeds allowable perturbation limits, thus thwarting attacks. Compactness of normal samples further enhances robustness by reducing intra-class variability, ensuring that small perturbations are unlikely to push normal samples into the anomaly space. Together, these properties strengthen the model’s decision boundaries and reduce overlap between classes, minimizing both false positives and negatives under adversarial conditions.
>
> This separation improves generalization by enabling the model to learn distinct and discriminative features, making it resilient to unseen samples and adversarial manipulations. Additionally, robust feature learning ensures the model focuses on essential characteristics that differentiate normal and anomaly samples, further bolstering resistance to attacks.
>
> In our method, this principle is implemented through the construction of opposite pairs between normal samples and their pseudo-anomalies. Our COBRA loss function explicitly pulls positive pairs together to ensure compactness while pushing apart opposite pairs to maximize the margin. Adversarial training then reinforces this separation by generating perturbations that directly target the inter-group boundary, enhancing robustness against attacks near the decision threshold.
>
> Notably, the ablation studies (Table 6) and the visualization (Figure 3) provide supportive evidence to these claims
>
>  >**Q6:**
>
> *We will work on refining the manuscript to ensure these concepts are presented more clearly before the final submission. For further clarification, we have provided a brief explanation here:*
>
>
> In our setup where only normal samples are available during training, pseudo-anomalies are generated by applying **hard transformations** (e.g., blurring, intense cropping, random erasing transformations) to normal samples. For a normal sample $x$, the transformed version $x'$ is created as:
>
> $x' = T_m(\ldots T_2(T_1(x)))$
>
> where $T_i \in T$ is a transformation, and $m$ is the number of transformations applied.
>
>
> To ensure $x'$ sufficiently deviates from normal samples, a **thresholding mechanism** is applied:
> - **Feature Extraction**: Embeddings are obtained using a pre-trained model $C$.
> - **Likelihood Estimation**: A statistical model (e.g., Gaussian Mixture Model) fits the embeddings of normal samples.
> - **Threshold ($\lambda$)**: Based on a significance level, samples with a likelihood below $\lambda$ are accepted as pseudo-anomalies.
>
>
>
> An **opposite pair** consists of a normal sample and its corresponding pseudo-anomaly:
>
> - **Opposite Pair** $(x, x')$: The normal sample $x$ and the transformed sample $x'$ that is accepted as a pseudo-anomaly.
>
>
> The purpose of creating negative pairs is to mitigate the inclusion of opposite pairs and maximize the margin between normal and anomaly samples, ensuring greater separation in the feature space for robust anomaly detection.
>
>
>
>
>
>
>
>
> In the contrastive learning framework, for any given normal sample, positives and negatives are defined as follows:
>
>   Positive Pairs
>
> - **Definition**: Pairs of augmented versions of the same sample from the **same group** (either both normal or both pseudo-anomalies).
> - **Construction**:
>   - Apply light augmentations $\tau_1$ and $\tau_2$ to the normal sample $x$ to obtain $x_1$ and $x_2$.
>   - **Positive Pair**: $(x_1, x_2)$.
> - **Purpose**: Encourage the model to learn invariant features within the same class, promoting compactness.
>
>
>  Negative Pairs
>
> - **Standard Approach**: Traditionally, negatives are all other samples in the batch, which can include samples from the same class, leading to **spurious negative pairs** (e.g., normal-normal pairs).
> - **Our Approach**:
>   - **Opposite Pairs as Negatives**: Specifically use opposite pairs $(x, x')$ as negative pairs.
>   - **Definition**: Negative pairs are formed between a normal sample and its corresponding pseudo-anomaly.
> - **Purpose**: Focus on maximizing the margin between normal and anomaly samples without being misled by spurious negatives.
>
>
>  >**Q7:**
>
> Thank you for your observation. The loss formulation indeed aligns with NT-Xent as introduced in SimCLR. We have revised the manuscript and updated the terminology and references accordingly to reflect this.
>
>  >**Q8&W3&W4:**
>
> Thank you for the feedback. We will revise the manuscript to align with the proposed suggestions and enhance the clarity of the definitions. We appreciate your input and will apply it to improve the manuscript.

---

> > ### Comment · Reviewer_YXiZ · 2024-11-26
> >
> > Thank you for the detailed and thoughtful response. Your clarifications and additional results address my concerns effectively. I will maintain my score.

---

> > > ### Author Response · Authors · 2024-11-27
> > >
> > > Dear Reviewer YXiZ,
> > >
> > > Thank you for your review and for taking the time to consider our responses. We are glad that your concerns have been addressed.
> > >
> > > Sincerely, The Authors

---

### Meta-Review · Area_Chair_Uowj · 2024-12-19

**Metareview:**

Based on the reviews, I recommend accepting the paper. The paper has received four high-quality reviews, all of which recommend acceptance. The reviewers highlight several strengths of the work, including the thorough empirical evaluation.

**Additional Comments On Reviewer Discussion:**

The discussion centered around the clarity of the paper's methodology, terminology, and empirical analysis, with most reviewers initially highlighting concerns regarding thefollowing aspects:
- **Reviewer YXiZ:** Raised concerns about the effectiveness of mitigating spurious negative pairs and issues with terminology and presentation.
- **Reviewer 1DsR:** Pointed out unclear or inconsistent content and a lack of sufficient analysis to explain the experimental results.
- **Reviewers qBmY &  mZoT:** Both reviewers found the paper lacking in detailed explanations and clear empirical analysis.

The authors provided rigorous and detailed responses to all concerns, including additional results, clearer explanations, and a commitment to revise the manuscript to improve clarity. As a result, all reviewers found the authors' responses satisfactory, and concerns were effectively addressed.

In weighing these points, the key concerns around clarity and empirical support were resolved to the satisfaction of the reviewers, justifying the final decision to recommend acceptance.

---

### Decision · Program_Chairs · 2025-01-22

Accept (Poster)